# Measurement Report: Effects of anthropogenic emissions and environmental factors on biogenic secondary organic aerosol (BSOA) formation in a coastal city of Southeastern China

Youwei Hong[a,b,c,d*], Xinbei Xu[a,b,c], Dan Liao[e], Taotao Liu[a,b,c],Xiaoting Ji[a,b,c], Ke Xu[a,b,d], Chunyang Liao[f], Ting Wang[g], Chunshui Lin[g], Jinsheng Chen[a,b,c*]

[a]Center for Excellence in Regional Atmospheric Environment, Institute of Urban Environment, Chinese Academy of Sciences, Xiamen, 361021, China

[b]Key Lab of Urban Environment and Health, Institute of Urban Environment, Chinese Academy of Sciences, Xiamen, 361021, China

[c]University of Chinese Academy of Sciences, Beijing, 100049, China

[d]School of Life Sciences, Hebei University, Baoding, 071000, China

[e]College of Environment and Public Health, Xiamen Huaxia University, Xiamen 361024, China

[f]State Key Laboratory of Environmental Chemistry and Ecotoxicology, Research Center for Eco-Environmental Sciences, Chinese Academy of Sciences, Beijing 100085, China

[g] Institute of Earth Environment, Chinese Academy of Sciences, Xi'an, 710061, China

*Corresponding author E-mail: Jinsheng Chen (jschen@iue.ac.cn); Youwei Hong (ywhong@iue.ac.cn)

**Abstract:**

To better understand the formation of biogenic secondary organic aerosol (BSOA), aerosol samples with a 4 h time resolution were collected during summer and wintertime in the southeast of China, along with on-line measurements of trace gases, aerosol chemical compositions, and meteorological parameters. The samples were analyzed by gas chromatography-mass spectrometry for $PM_{2.5}$-bound SOA tracers, including isoprene ($SOA_I$), α/β-pinene ($SOA_M$), β-caryophyllene ($SOA_C$), and toluene (ASOA). The average concentrations of total SOA tracers in winter and summer were 38.8 and 111.9 ng $m^{-3}$, respectively, with the predominance of $SOA_M$ (70.1% and 45.8%), followed by $SOA_I$ (14.0% and 45.6%), ASOA (11.0% and 6.2%) and $SOA_C$ (4.9% and 2.3%). Compare to those in winter, the majority of BSOA tracers in summer showed significant positive correlations with Ox ($O_3$+$NO_2$), HONO, ultraviolet (UV) and temperature (T), indicating the influence of photochemical oxidation under relatively clean conditions. However, in winter, BSOA tracers were significantly correlated with $PM_{2.5}$, $NO_3^-$, $SO_4^{2-}$, and $NH_3$, attributed to the contributions of anthropogenic emissions. Major BSOA tracers in both seasons was linearly correlated with aerosol acidity (pH), liquid water content (LWC) and $SO_4^{2-}$. The results indicated that acid-catalyzed reactive uptake onto sulfate aerosol particles enhanced the formation of BSOA. In summer, the clean air mass originated from the ocean, and chlorine depletion was observed. We also found that concentrations of the total SOA tracers was correlated with HCl and chlorine ions in $PM_{2.5}$, reflecting the contribution of Cl-initiated VOC oxidations to the formation of SOA. In winter, the northeast dominant wind direction brought continental polluted air mass to the monitoring site, affecting the transformation of BSOA tracers. This implied that anthropogenic emissions, atmospheric oxidation capacity and halogen chemistry have significant effects on the formation of BSOA in the southeast coastal area.

**Keywords:** SOA tracers; biogenic volatile organic compounds; anthropogenic pollutants; atmospheric oxidation capacity; coastal area

# 1. Introduction

Secondary organic aerosol (SOA) has attracted widespread scientific researchers concerns, due to its potential impacts on climate change, human health and air quality (Shrivastava et al., 2017; Reid et al., 2018; Zhu et al., 2019; Wang et al., 2021b). Understanding the formation of SOA and assessing its relevance for environmental effects become an integral part of aerosol chemistry (Charan et al., 2019; Xiao et al., 2020; Palmer et al., 2022). However, due to its complex precursors and atmospheric physical or chemical processes, SOA prediction by air quality models remains highly uncertain (McFiggans et al., 2019). Therefore, it is necessary to better explore missed SOA sources and unknown SOA formation mechanisms.

SOA was produced by the conversion of biogenic and anthropogenic volatile organic compounds (BVOCs and AVOCs) through complex homogeneous and heterogeneous reactions (Charan et al., 2019; Xiao et al., 2020; Mahilang et al., 2021). BVOCs are the main precursors of SOA on a global scale, while AVOCs are the predominant contributor to SOA in urban areas (Hallquist et al., 2009; Wang et al., 2021a). Recently, laboratory, field observation and model studies have shown that anthropogenic emissions greatly affect the formation of BSOA (Hoyle et al., 2011; Shrivastava et al., 2019; Zhang et al., 2019b; Zhang et al., 2019c; Mahilang et al., 2021; Xu et al., 2021). Anthropogenic air pollutants, such as NOx, $SO_2$, $NH_3$ and aerosols, could influence the conversion of BVOCs to the particulate phase and the production of nitrogen and sulfur compounds (Wang et al., 2020). NOx is one of the important drivers of SOA formation and yields during both daytime and nighttime through alternating the fate of peroxy radicals ($RO_2\cdot$) (Sarrafzadeh et al., 2016; Newland et al., 2021). While ·OH dominates the photochemical oxidation of BVOC during daylight hours, and $NO_3\cdot$becomes one of the main oxidants for biogenic SOA and organic nitrates at night. $SO_2$ also plays an important role in changing SOA formation from BVOC photooxidation and ozonolysis through sulfuric acid formation and acid-catalyzed heterogeneous reactions (Zhao et al., 2018; Zhang et al., 2019b; Xu et al.,

2021). In addition, $NH_3$ and amines can affect the SOA yields and composition through both gas-phase and heterogeneous reactions, by reacting with sulfuric or nitric acid to generate secondary inorganic aerosols (SIA) (Ma et al., 2018; Liu et al., 2021; Lv et al.,2022). However, due to complex precursors and atmospheric processes, the combined effects of anthropogenic emissions and meteorological factors on the formation of SOA are not fully understood.

The coastal area of southeastern China is under the East Asian monsoon control, which cause an obvious alternation of polluted and clean air masses from continental and ocean area, respectively (Wu et al., 2019; Hong et al., 2021). Also, the local geographical environment, including relatively high humidity, dense vegetation and strong atmospheric oxidation capacity, provides a good chance to study the sources and formation mechanisms of SOA. In our previous studies, ground-based observations in a mountainous forest area of this region showed that BSOA tracers were the largest contributor to SOA, and the aerosols were highly oxidized (Hong et al., 2019). However, with the development of rapid urbanization, anthropogenic emissions will be of great significance on SOA formation (Liu et al., 2020). Halogen radicals (chlorine, bromine, iodine) have an important role in tropospheric oxidants chemistry and OA formation (Wang et al., 2021c). Therefore, it is necessary to investigate the sources and formation mechanisms of SOA in coastal urban areas, and so as to provide a scientific basis for the estimation of regional SOA budgets and $PM_{2.5}$ pollution control.

In this study, a continuous $PM_{2.5}$ sampling campaign with a 4 h time resolution was conducted in a coastal city of southeastern China during the winter and summertime period. Seasonal, diurnal variations and SOC contributions of SOA tracers were analyzed. We also demonstrated the indications of SOA tracers for air pollution process. Finally, the combined effects of anthropogenic emissions and major environmental factors on promoting SOA formation was discussed.

## 2. Materials and methods

*2.1 Sample collection*

The sampling was performed at the Institute of Urban Environment, Chinese Academy of Sciences (118.06° E, 24.61° N), which is located in a suburban area of Xiamen, a coastal city of southeastern China. Detailed information of the air monitoring supersite was described in our previous study (Hong et al., 2021). Briefly, time-resolved (00:00–08:00, 08:00–12:00, 12:00–16:00, 16:00–20:00, 20:00–24:00 CST – China Standard Time) $PM_{2.5}$ samples were collected on the rooftop of the station (about 70m above the ground). The sampling was carried out by using a high volume (1.05 $m^3$ $min^{-1}$) sampler (TH-1000C, Wuhan Tianhong, China) with a $PM_{2.5}$ inlet from 10 to 18 January, and from 5 to 14 July 2020. All samples were collected onto pre-baked (450 ◦C, 6 h) quartz fiber filters. Field blank samples were also collected. The sample filters were separately sealed in aluminum foil and stored in a freezer (−20 ◦C) prior to analysis.

*2.2 SOA tracers analysis by GC/MS*

The isoprene-derived SOA ($SOA_I$) tracers included 2 methyltetrols (MTLs: 2-methylthreitol (MTL1) and 2-methylerythritol (MTL2)), C5-alkene triols (cis-2-methyl-1,3,4-trihydroxy-1-butene, trans-2-methyl-1,3,4-trihydroxy-1-butene, and 3-methyl-2,3,4-trihydroxy-1-butene) and 2-methylglyceric acid (MGA). The monoterpene-derived SOA ($SOA_M$) tracers were composed of pinic acid (PA), pinonic acid (PNA), 3-hydroxyglutaric acid (HGA), 3-methyl-1,2,3-butanetricarboxylic acid (MBTCA), 3-hydro-4,4-dimethyglutaric acid (HDMGA), and 3-acetylglutaric acid (AGA). The β-caryophyllene-derived SOA ($SOA_C$) tracer was β-caryophyllenic acid (CA), the toluene-derived SOA ($SOA_A$) tracer was 2,3-Dihydroxy-4-oxopentanoic acid (DHOPA) and levoglucosan (LEV) as a tracer of biomass burning. Due to the lack of authentic standards, surrogate standards (including erythritol, malic acid, PA and citramalic acid) were used to quantify $SOA_I$, $SOA_M$, $SOA_C$ and $SOA_A$ tracer, respectively (Fu et al., 2009). Details of SOA tracer's calculated concentrations based on relative response factors (RRFs) were presented in our previous studies (Hong et al., 2019; Liu et al., 2020).

The analytical procedure of fifteen SOA tracers was published in our previous studies (Hong et al., 2019; Liu et al., 2020). Briefly, the filter samples were

ultrasonically extracted with a mixture of dichloromethane and methanol (2:1, v/v) for 10 min. The extracts were filtered with a PTFE filter (0.22 μm), and dried with high purity $N_2$ (99.99%), and then derivatized with 60 μL of N,O-bis-(trimethylsilyl) trifluoroacetamide (BSTFA) with 1% trimethylsilyl chloride and 10 μL of pyridine at 70 °C for 3 h. At last, 140 μL of internal standard solution ($^{13}C$ n-alkane solution, 1.507 ng μ $L^{-1}$) was added into the samples.

Fifteen SOA tracers were determined by GC-MSD (7890A/5975C, Agilent Technologies, Inc., USA) with a DB-5 MS silica capillary column (i.d. 30×0.25 mm, 0.25 μm film thickness). 1 μL sample was injected with splitless mode and high purity helium (99.999%) was used as carrier gas at a stable flow of 1.0 mL/min. The GC temperature was initiated at 100 °C (held for 1 min) and then to 300 °C at 5 °C $min^{-1}$, and kept at 300 °C for 10 min. The operation mode is electron ionization (EI) mode of 70 ev. The method detection limits (MDLs) for erythritol and PNA were 0.01 and 0.02 ng $m^{-3}$, respectively. The recoveries of erythritol, PNA, malic acid, PA and citramalic acid were 67±2%, 73±1%, 75±1%, 88±7% and 82±8%, respectively. SOA tracers were not detected in the field blank samples.

*2.3 Observations in the air monitoring supersite*

Water-soluble inorganic ions (WSII) in $PM_{2.5}$ ($Cl^-$, $SO_4^{2-}$, $NO_3^-$, $Na^+$, $K^+$, $NH_4^+$, $Mg^{2+}$, and $Ca^{2+}$) and gas pollutants (HCl, HONO, $HNO_3$, $NH_3$) were hourly measured using a monitoring device for aerosols and gases in ambient Air (MARGA 2080; Metrohm Applikon B.V.; Delft, Netherlands). Internal calibration was carried out using LiBr standard solutions. The detection limit of $Cl^-$, $SO_4^{2-}$, $NO_3^-$, $Na^+$, $K^+$, $NH_4^+$, $Mg^{2+}$, and $Ca^{2+}$ were 0.01, 0.04, 0.05, 0.05, 0.09, 0.05, 0.06 and 0.09 μg $m^{-3}$, respectively.

Hourly mass concentrations of $PM_{2.5}$ and $PM_{10}$ were measured by using a tapered element oscillating microbalance (TEOM1405, Thermo Scientific Corp., MA, USA). $NO_2$, $SO_2$, and $O_3$ were monitored using continuous gas analyzers (TEI 42i, 43i, and 49i, Thermo Scientific Corp., MA, USA). Ambient meteorological parameters including relative humidity (RH), temperature (T), wind speed (WS), and wind direction (WD) were obtained by an ultrasonic atmospherium (150WX, Airmar, the

USA). Photolysis frequencies were determined using a photolysis spectrometer (PFS-100, Focused Photonics Inc., Hangzhou, China), including the photolysis rate constants *J* ($O^1D$), *J* (HCHO_M), *J* (HCHO_R), *J* ($NO_2$), *J* ($H_2O_2$), *J* (HONO), *J* ($NO_3$_M) and *J* ($NO_3$_R), and the spectral band ranged from 270 to 790 nm. Boundary layer height (BLH) based on ERA-5 reanalysis dataset was downloaded from the following link https://www.ecmwf.int/en/forecasts/datasets/reanalysis-datasets/era5.

### *2.4 Estimation of SOC using a tracer-based method*

The fraction of SOC formed by the oxidation of monoterpene, isoprene, β-caryophyllene and toluene was estimated using a tracer-based method (Kleindienst et al., 2007; Hong et al., 2019). It is defined as [SOC] = $\sum$ i[tri]/$f_{SOC}$, where [SOC] represents the mass concentration of SOC (μgC $m^{-3}$) and $\sum$ i [tri] means the sum of the concentration of individual SOA tracer (μg $m^{-3}$). The carbon mass fractions ($f_{SOC}$) of monoterpene, isoprene, β-caryophyllene and toluene were $0.231 \pm 0.111$, $0.155 \pm 0.111$, $0.023 \pm 0.005$ and $0.008 \pm 0.003$, respectively, based on smog-chamber experimental data (Kleindienst et al., 2007).

### *2.5 Aerosol acidity and OH calculation*

The E-AIM IV (Extended Aerosol Inorganic Model IV version) was used to simulate the aqueous and solid phases of ionic compositions in the mixing system ($H^+$-$NH_4^+$-$SO_4^{2-}$-$NO_3^-$-$Cl^-$-$Na^+$-$H_2O$) at a given T and RH (Friese and Ebel, 2010). According to our previous study (Wu et al., 2020), the hourly averaged T, RH, $SO_4^{2-}$, $NO_3^-$, $Cl^-$, $NH_4^+$, $Na^+$ and molar concentrations of total aerosol acidity ($H^+_{total}$) were used as the input in the model E-AIMIV to obtain the concentrations of free ions (including free $H^+$ ($H^+_{insitu}$) in the aqueous phase, and liquid water content (LWC)) . $H^+_{insitu}$ defined as the moles of free hydrogen ions in the aqueous phase of aerosols per unit of air (nmol·$m^{-3}$), is the actual acidity in the droplets of the aerosols. The $H^+_{total}$ was estimated from the ionic balance of the relevant ionic species: $H^+_{total} = 2SO_4^{2-}+NO_3^-+Cl^--NH_4^+-Na^+$.

The pH of aerosol was calculated as the follow:

$$pH = -\lg\left(\frac{\gamma \times H^{+}_{insitu}}{V_{aq}/1000}\right)$$

where γ and $V_{aq}$ denote the activity coefficient and the volume of particle aqueous phase in air ($cm^3 \cdot m$).

The forward mode of ISORROPIA II thermodynamic model was run by assuming that aerosol solutions were metastable (only a liquid phase). The pH value from ISORROPIA II was calculated using the following equation:

$$pH = -\lg\left(\frac{1000 \times H^{+}}{LWC}\right)$$

where $H^+$ is the equilibrium particle hydronium ion concentration per volume air.

The OH concentration ([OH]) was estimated using the $NO_2$ and HONO concentrations and the photolysis rate constants (J) of $NO_2$, $O_3$, and HONO, according to the following improved empirical formula (Wen et al., 2019).

$$[OH] = 4.1 \times 10^9 \times \frac{J(O^1D)^{0.83} \times J(NO_2)^{0.19} \times (140 \times NO_2 + 1) + HONO \times J(HONO)}{0.41 \times NO_2^2 + 1.7 \times NO_2 + 1 + NO \times k_{NO+OH} + HONO \times k_{HONO+OH}}$$

### *2.6 Statistical analysis*

Correlation analysis by SPSS 22.0 software (IBM, Armonk, NY, USA) was used to study the relationship among SOA tracers, meteorological parameters and criteria air pollutants. One-way analysis of variance (ANOVA) was adopted to examine the variations of different factors.

### 2.7. Backward trajectory analysis

Hybrid Single-Particle Lagrangian Integrated Trajectory (HYSPLIT) was used to analyze the impacts of air masses on Xiamen during different seasons. 72 h backward trajectories were calculated every hour at a height of 500 m. The meteorological data with a resolution of 1° longitude × 1° latitude was obtained from the NCEP/GDAS. Cluster analysis was adopted using the total spatial variance (TSV).

## 3 Results and discussion

### *3.1. Overview of air pollutants*

The concentrations of criteria air pollutants, including $SO_2$, CO, $NO_2$, $O_3$, $PM_{2.5}$

and $PM_{10}$, and meteorological parameters during wintertime and summertime were shown in Fig.1. The concentrations of $PM_{2.5}$ in winter ranged from 14.9 to 75.3 μg m$^{-3}$ with an average of 42.1 μg m$^{-3}$, which was much higher than that (the average of 18.4 μg m$^{-3}$) in summer, ranging from 12.8 to 46.4 μg m$^{-3}$. The concentrations of CO, $NO_2$ and $PM_{10}$ showed similar seasonal trends to the pattern of $PM_{2.5}$. In contrast, $O_3$ had the highest concentration in summer, which was attributed to the formation of photochemical reaction under strong UV radiation and the weak titration of nitrogen oxides. Meanwhile, the concentrations of $SO_2$ (8.37±0.79 μg m$^{-3}$) in summer was also higher than that (2.63±1.95 μg m$^{-3}$) in winter, mainly attributed to the influence of coal combustion and ship emissions. The monitoring site was located approximately 15 km away from Xiamen port area and a coal-fired power plant (4 × 300 kW) in the south. Southerly winds were prevailed in summer, which might cause the relative high concentration of $SO_2$ in the monitoring site.

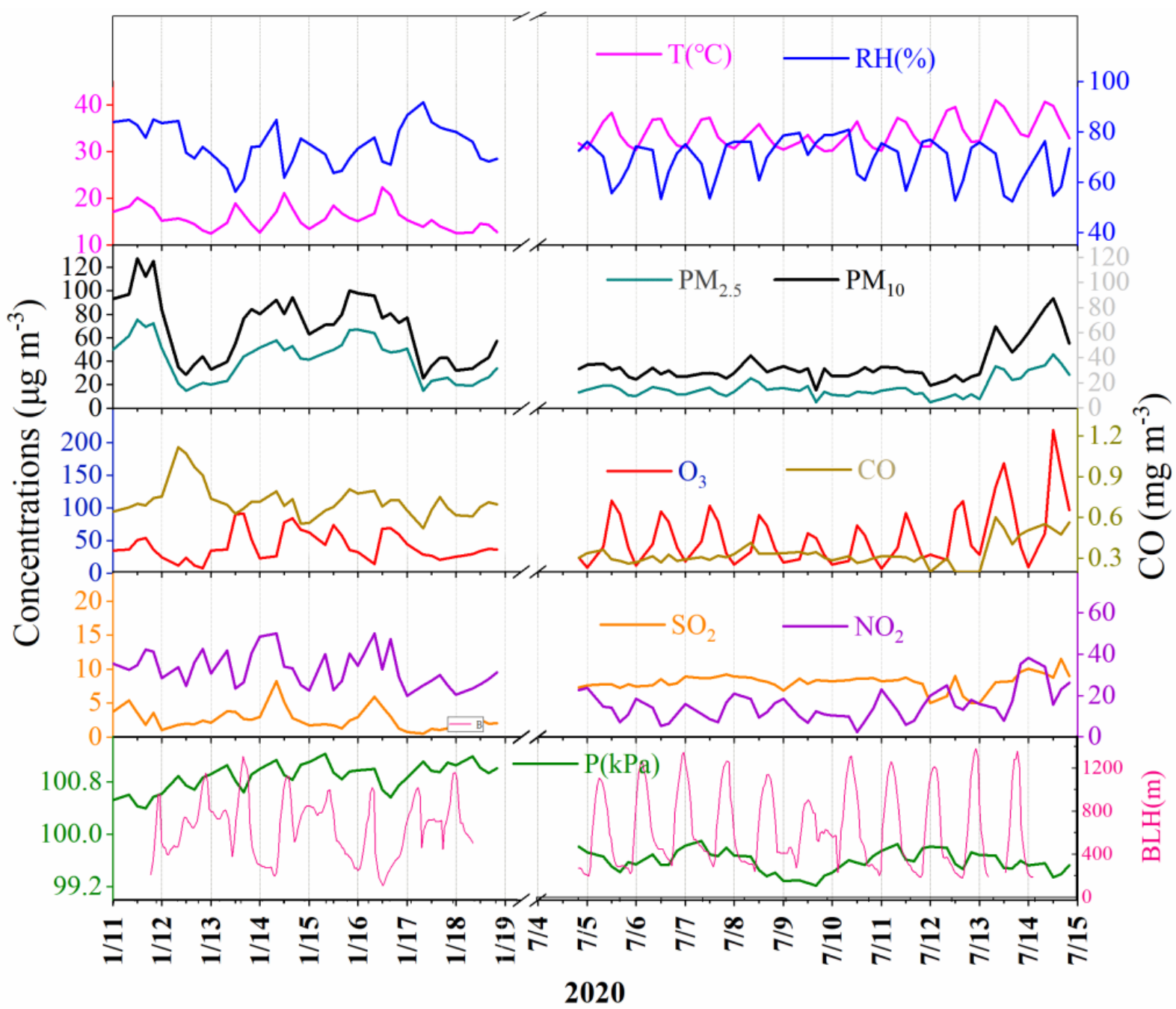

**Figure 1. Time series of criteria air pollutants and meteorological parameters during the sampling period**

*3.2 Temporal variations of SOA tracers* and estimated *SOC*

Temporal variations of individual SOA tracer are shown in Fig.S1. The average concentrations of total SOA tracers in winter and summer were 38.8 and 111.9 ng $m^{-3}$, respectively, with the predominance of $SOA_M$, followed by $SOA_I$ and $SOA_C$. In summer, BSOA tracers showed much higher concentrations in the daytime than in the nighttime, while inverse results were observed in winter. For example, in summer, $SOA_I$ in the daytime ranged from 21.3 to 293.2 ng $m^{-3}$ (average of 82.6±65.3ng $m^{-3}$) and the concentrations of $SOA_I$ ranging from 6.81 to 110.1 ng $m^{-3}$ (average of 27.4±24.6 ng $m^{-3}$) were observed in the nighttime. However, in winter, the concentrations of isoprene SOA tracers in the daytime ranging from 1.36 to 11.1 ng $m^{-3}$ (average of 4.93±2.62ng $m^{-3}$) were lower than those (average of 15.3±8.32 ng $m^{-3}$) in the nighttime. As shown in Fig. 2, diurnal variations of $SOA_M$, $SOA_I$, CPA and DHOPA tracers in summer showed high levels in the afternoon (12:00–16:00 CST), due to the impacts of beneficial photochemical oxidation conditions caused by high temperature and strong UV radiation. The related SOA tracers were consisted with the emissions of their precursors including biogenic and anthropogenic VOCs, similar to our previous studies (Hong et al., 2019; Liu et al., 2020). However, the SOA tracers in winter showed the lowest concentrations in the morning (8:00–12:00 CST), related with the favorable dispersion conditions caused by the increasing planetary boundary layer height (BLH) (Fig.1). Totally, high concentrations of BSOA tracers was found in the daytime and in summer, indicating the effects of temperature on biogenic VOCs emissions and their photochemical oxidations. And the concentrations of BSOA tracers in winter increased in the nighttime, due to the changing of nocturnal boundary layer.

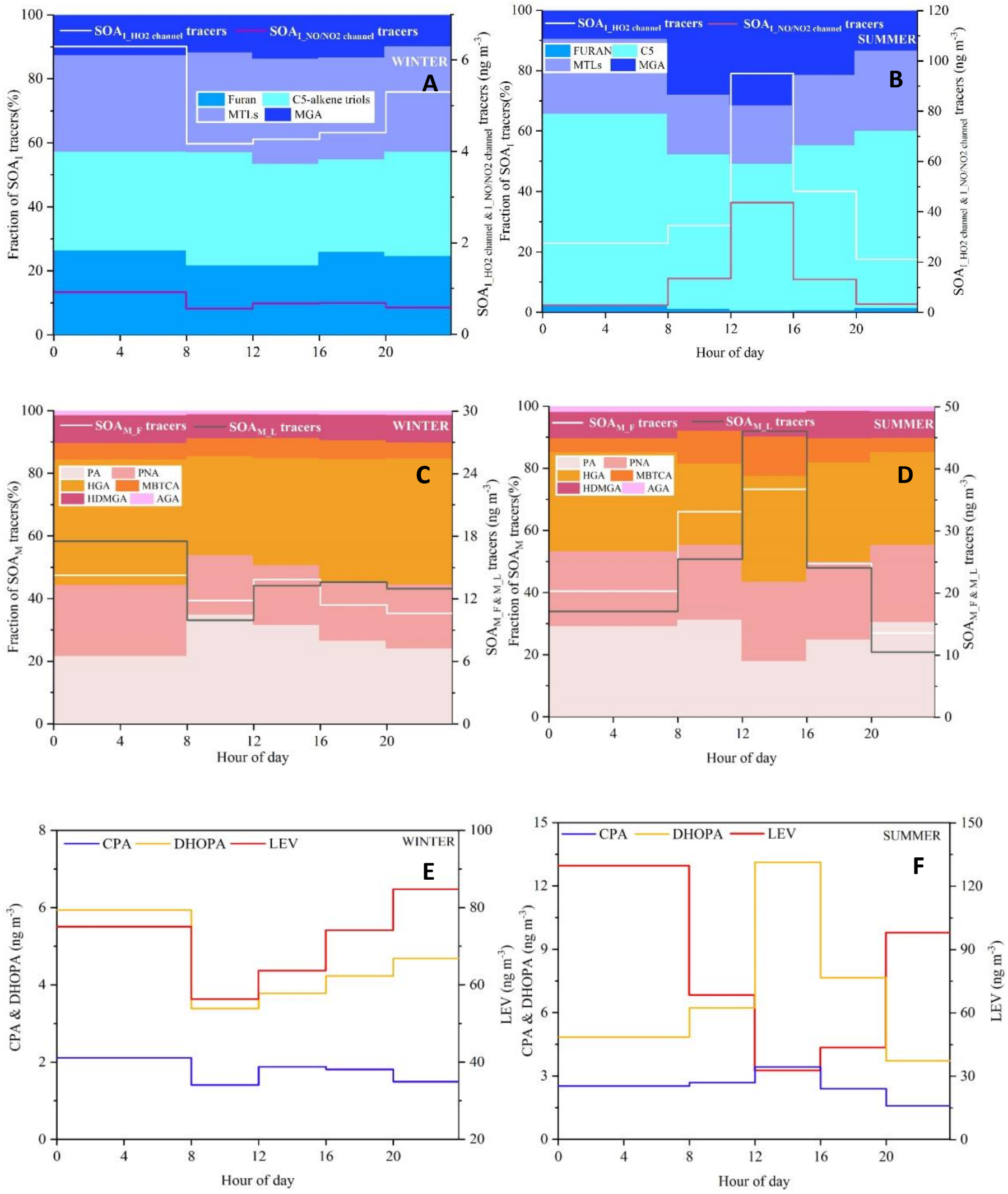

**Figure 2. Diurnal variation of individual SOA tracer during the wintertime and summertime**

As shown in Fig.3a, b, SOA tracers-based SOC in winter and summer was estimated. The concentrations of SOC in winter ranged from 0.27 to 2.36 μg C m$^{-3}$, with an average of 1.11 μg C m$^{-3}$. Meanwhile, the concentrations of SOC in summer ranged from 0.46 to 7.85 μg C m$^{-3}$, with an average of 2.27 μg C m$^{-3}$. The results showed that the contributions of SOA tracers to SOC in summer was higher than those in winter.

For individual SOA tracer, the concentrations of monoterpene-derived SOC (0.87±0.64 μg C m$^{-3}$) was comparable to the toluene-derived SOC (0.90±0.69 μg C m$^{-3}$), which were higher than isoprene-derived SOC (0.39 ± 0.38 μg C m$^{-3}$) and β-caryophyllene-derived SOC (0.10 ± 0.08 μg C m$^{-3}$). An obvious trend of diurnal variations of isoprene-derived SOC was observed, which was consistent with a certain amount of isoprene emitted from various plants. However, no similar trend was found in winter. In addition, the toluene, monoterpene, isoprene and β-caryophyllene-derived SOC in summer accounted for 40.0%, 39.2%, 15.7% and 5.1% of the total SOC, respectively (Fig.3c, d). However, in winter, the percentages of toluene, monoterpene, isoprene and β-caryophyllene-derived SOC were 47.2%, 42.1%, 3.2% and 7.6%, respectively. The percentages of isoprene-derived SOC estimated from different precursors varied significantly among the seasons. High temperature enhanced the emissions of isoprene, and strong solar radiation favored the formation of isoprene SOA tracers, contributing to the highest isoprene-derived SOC percentage in summer (Ding et al., 2014). And the highest percentages of toluene-derived SOC (47.2%) in winter were related with anthropogenic emissions and adverse diffusion conditions.

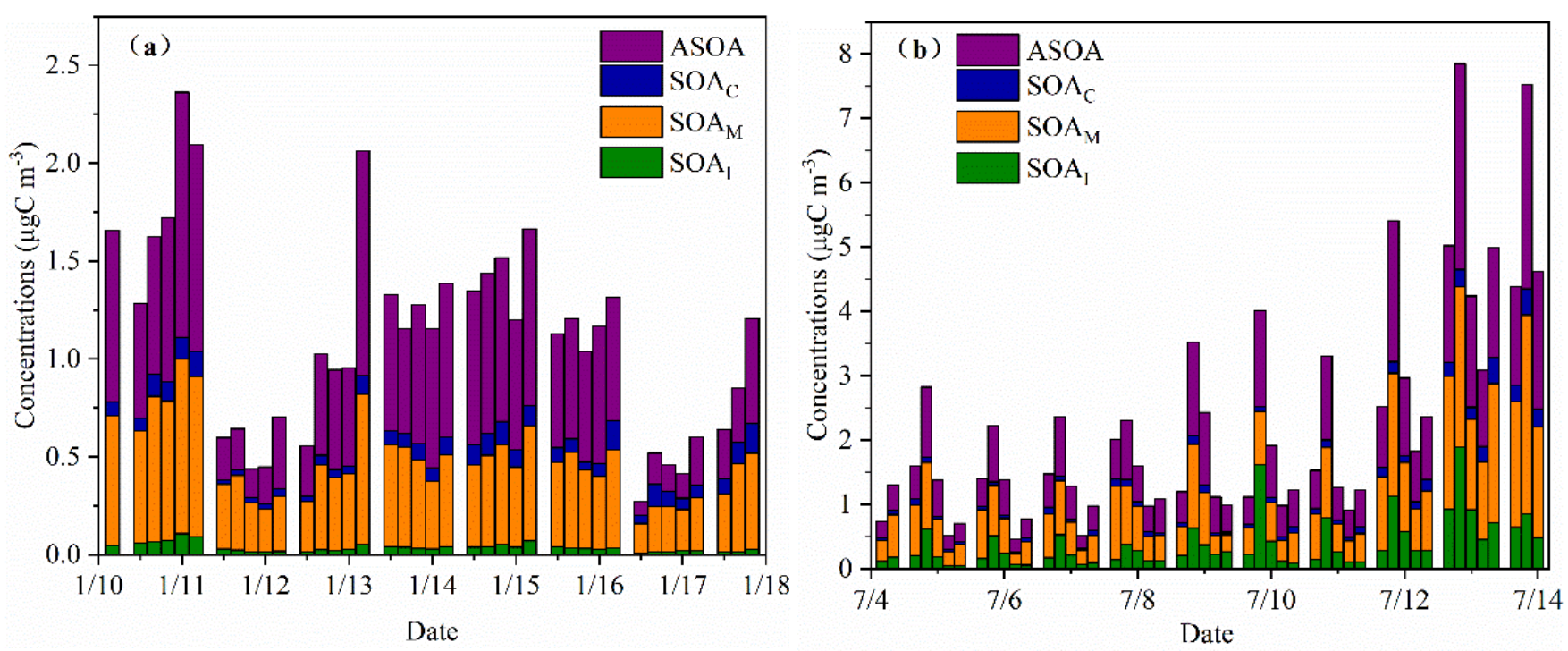

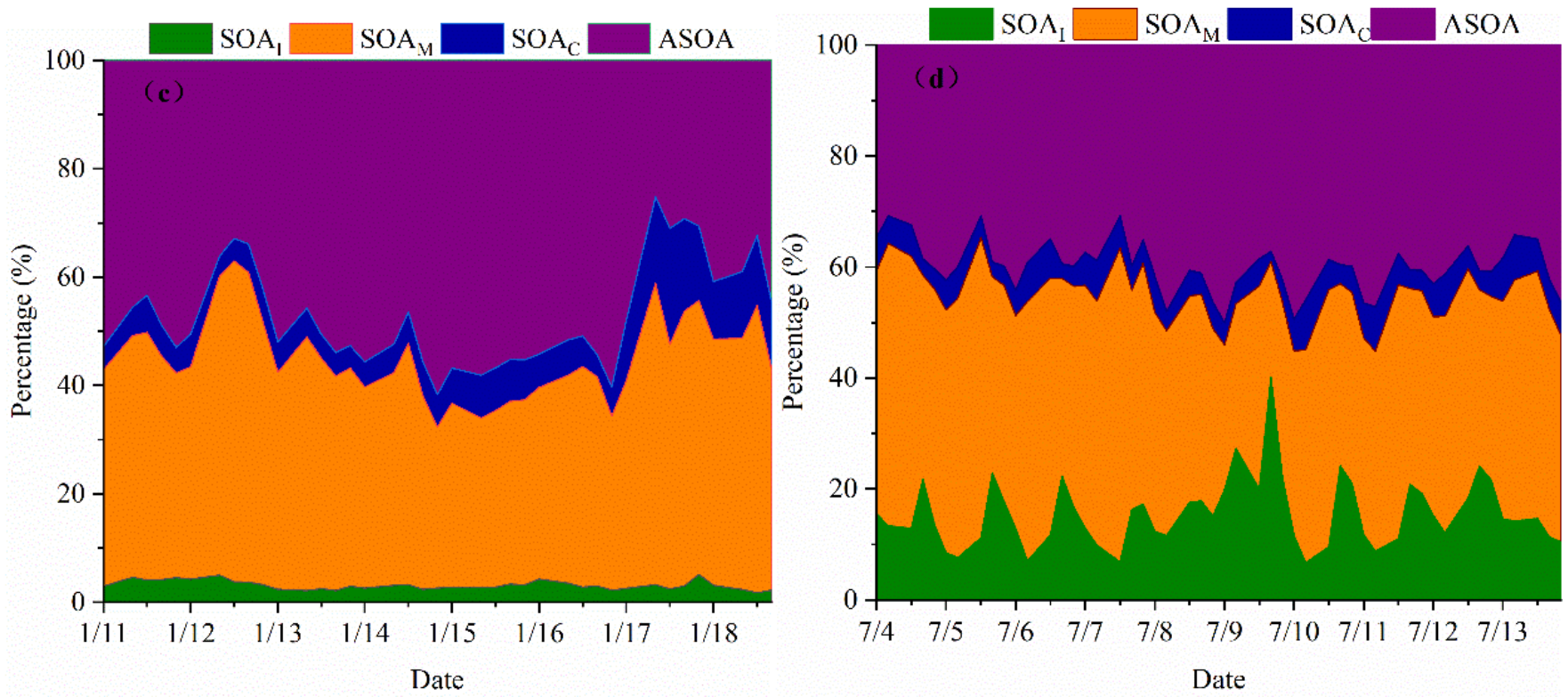

**Figure 3. Concentrations and percentages of SOA tracer-based estimated SOC during the sampling period**

*3.3 Atmospheric process indication of BSOA tracers*

As shown in Fig.4, percentages of different types of SOA tracers in winter and summer were calculated. In summer, the monoterpene, isoprene, toluene and β-caryophyllene SOA tracers accounted for 45.8%, 45.6%, 6.2% and 2.3% of the total SOA tracers, respectively. However, in winter, the percentages of monoterpene, isoprene, toluene and β-caryophyllene SOA tracers were 70.1%, 14.0%, 11.0% and 4.9%, respectively. The percentage of $SOA_I$ decreased sharply, due to the impacts of temperature on isoprene emissions, which was consisted with our previous findings (Hong et al., 2019). Meanwhile, the concentrations of $SOA_M$ were the largest in both seasons, due to a large amount of monoterpene emissions from the related plant species. Xiamen, an international garden city, located in coastal area of southeastern China. Monoterpene, such as α/β-pinene, is mostly emitted by coniferous plant and most flowers and fruits, while isoprene originates from broad-leaved trees and deciduous plants (Ding et al., 2014; Shrivastava et al., 2017; Yang et al., 2021).

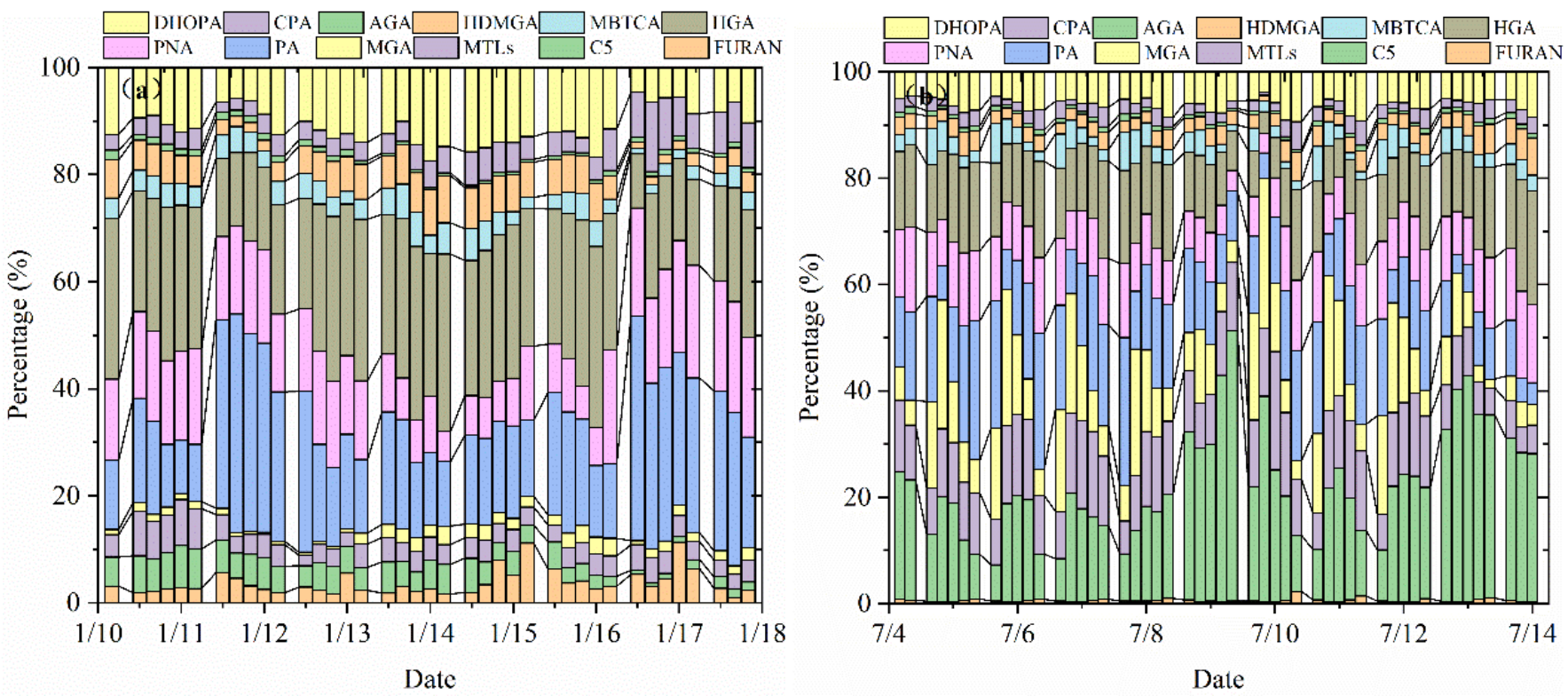

**Figure 4. Percentages of isoprene, monoterpene, β-caryophyllene and toluene SOA tracers in winter (a) and summer (b)**

The α/β-pinene SOA tracers, including first generation products (PA, PNA) and later-generation products (HGA, AGA, HDMGA and MBTCA), could be used to evaluate the aging degree of BSOA (Ding et al., 2014; Hong et al., 2019). In this study, HGA (32.2%) was the major component of α/β-pinene tracers, followed by PA (30.5%), PNA (21.8%), HDMGA (7.3%), MBTCA (6.8%), and AGA (1.5%). The percentage of PA and PNA were much higher than those in mountainous background areas (PA: 9% and PNA: 3%)(Hong et al., 2019), suggesting the contribution of preliminary products to SOA in urban areas. As shown in Fig.4, the percentages of PA and PNA in winter (21.8% and 14.2%) were higher than those in summer (14.2% and 10.7%). Reacted with atmospheric oxidants including $O_3$ and OH, PA and PNA were transformed into MBTCA (Offenberg et al., 2007). This is the reason why the proportions of PA and PNA had a significant decreasing trend from winter to summer. The ratio of MBTCA/(PA+PNA) in summer and winter were 0.16±0.09 and 0.12±0.07, respectively, which also proved the impacts of atmospheric oxidation capacity on the aging degree of $SOA_M$. In addition, the ratio of HGA/MBTCA could be used to distinguish the contribution of α-pinene or β-pinene to the $SOA_M$ formation (Jaoui et al., 2005; Ding et al., 2014). The ratio of HGA/MBTCA with an average of 5.78 in Xiamen was high, suggesting the contribution of β-pinene to $SOA_M$. Low ratio of HGA/MBTCA (~1.0)

showed that α-pinene was the major precursor for $SOA_M$ (Lewandowski et al., 2013).

As shown in Fig.4, MTLs and C5 alkene triols were the main components of the total $SOA_I$, with an average percentage of 68.0±14.9%, indicating a low-NOx environment (Ding et al., 2014; Liu et al., 2020). In summer, the percentages of MTLs and C5 alkene triols to the total SOA tracers in summer (21.8% and 14.2%) were obviously higher than those in winter (4.2% and 4.3%). This was consisted with the fact that the concentrations of $NO_2$ (14.8±7.46 μg $m^{-3}$) in summer was significantly lower than that (32.7±32.6 μg $m^{-3}$) in winter. Previous studies found that MTLs and C5 alkene triols were formed by the OH and $HO_2$ radicals via the $HO_2$ channel under low-NOx conditions (Surratt et al., 2010). C5 alkene triols are mainly produced by acid catalyzed reaction of Isoprene Epoxydiols (IEPOX) in the gas phase, while MTLs are formed by ring opening products of IEPOX (Surratt et al., 2007; Surratt et al., 2010). And the ozonolysis of isoprene was also an important pathway for MTLs in the presence of acid sulfate aerosols (Riva et al., 2016).

CPA, the typical tracer of sesquiterpenes, is formed by the photooxidation of β-caryophyllene (Jaoui et al., 2007). As shown in Fig.4, CPA in winter and summer accounted for 5.0% and 2.3% of the total SOA tracers, respectively. This is because that the percentage of $SOA_I$ has significant increase in summer. And the concentrations of CPA (2.5±2.0 ng $m^{-3}$) in summer were higher than that (1.7±0.8 ng $m^{-3}$) in winter, probably attributed to the emissions of β-caryophyllene driven by temperature and solar radiation. The CPA has a good correlation with DHOPA in summer (Fig.S2), suggesting the influence of photochemical oxidation (Liu et al., 2020). However, the CPA were not correlated with LEV in both seasons, reflecting the limited contribution of biomass burning (Zhang et al., 2019c).

### *3.4 Impacts of aerosol acidity on BSOA formation*

Aerosol acidity (pH) was an important factor on SOA formation (Surratt et al., 2007; Offenberg et al., 2009; Zhang et al., 2019b; Zhang et al., 2019d). Time series of aerosol pH calculated by ISORROPIA II is shown in Fig.5. The $PM_{2.5}$ in Xiamen was moderately acidic with daily pH range from 3.68 to 4.67. The highest aerosol pH was

observed in winter, and the lowest pH in summer. This is with similar seasonal trend, closing to the Yangtze River Delta (YRD) region, but obviously lower levels than those in NCP cities of China (Zhou et al., 2021). In general, the aerosol pH in Chinese cities were higher than those in US and European.

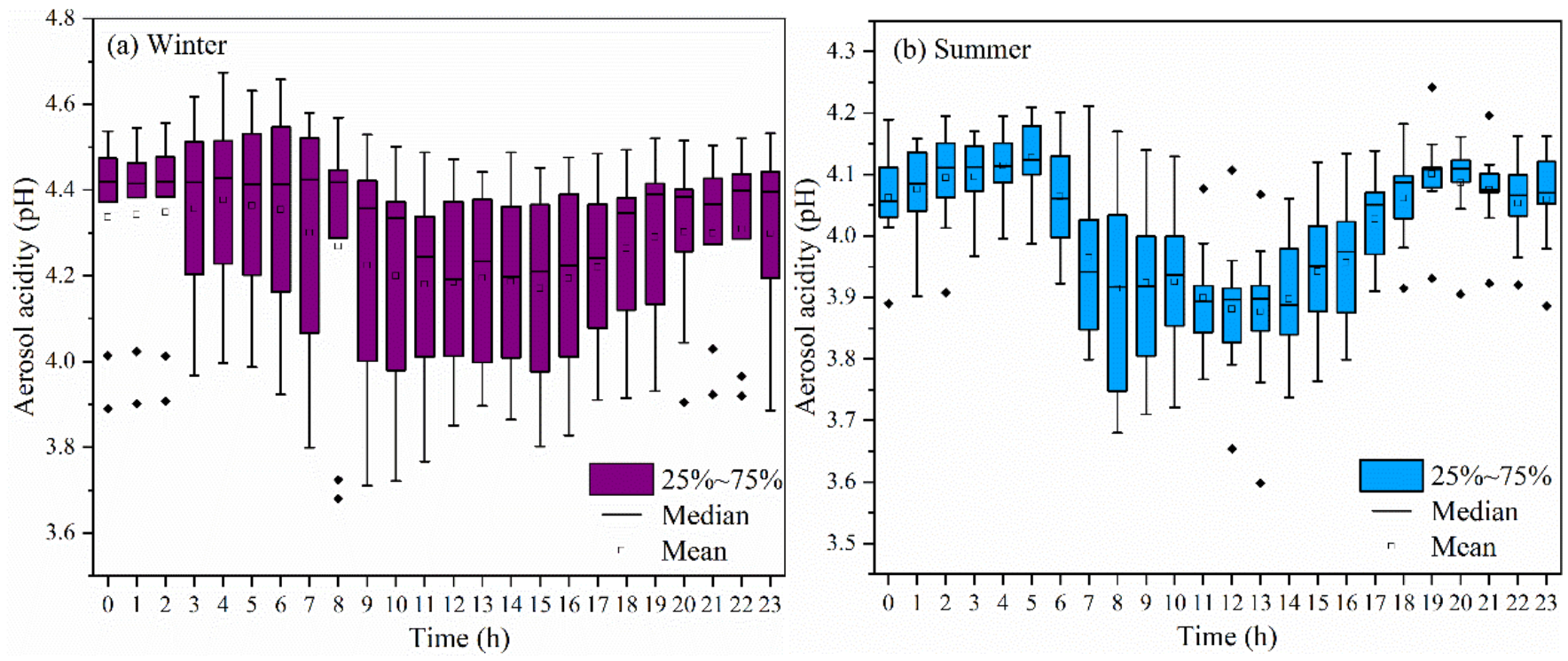

**Figure 5. Diurnal variations of aerosol acidity (pH) during the wintertime and summertime period (The boxes with error bars represent the 10th, 25th, 75th, and 90th percentiles)**

A declining trend pH during the daytime was observed (Fig. 5), which was related to the changes of chemical compositions and environmental conditions. The aerosol pH levels (~3 to 6) was related with a shift from sulfate- to nitrate-dominated aerosols (Guo et al., 2017). According to the multiphase buffer theory, the peak buffer pH (pKa*) regulated the aerosol pH, and temperature could obviously cause the variation of aerosol pH (Zheng et al., 2020). To further discuss the impacts of aerosol acidity on BSOA formation in coastal city, we analyzed the relationship between BSOA tracers and seed particles with different pH and liquid water content (LWC) (Fig. 6 and Table 1).

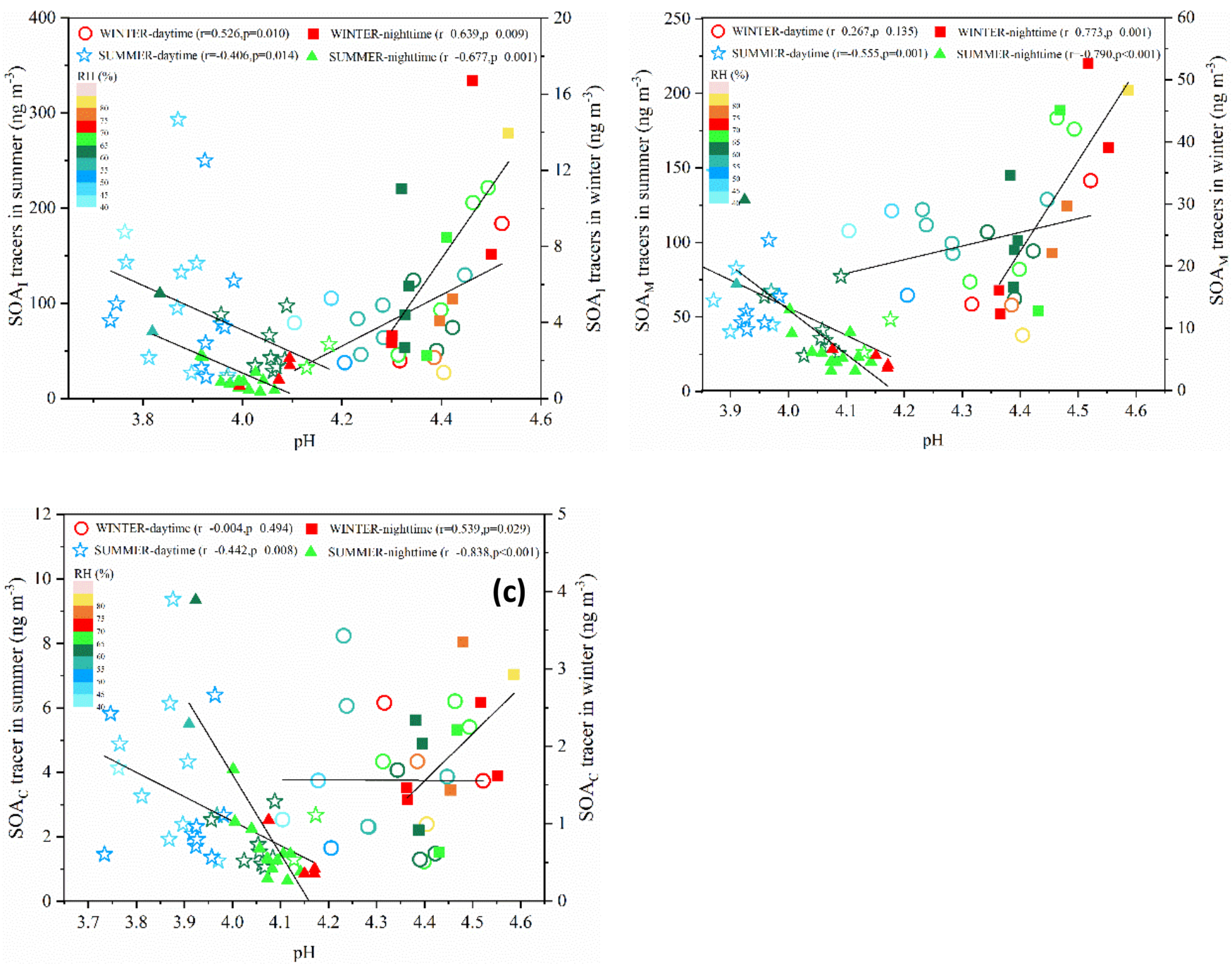

**Figure 6. Correlations of SOA$_I$ tracers (a), SOA$_M$ tracers (b), SOA$_C$ tracer (c) with aerosol acidity (pH) during the daytime and night-time**

In Table 1, the BSOA tracers was linearly correlated with aerosol acidity (pH) and $SO_4^{2-}$. In summer, BSOA tracers in the particle phase were found to increase with increasing acidity, which was attributed to the presence of acid catalyzed aerosols. For example, isoprene SOA tracers is mainly formed through acid-catalyzed reactive uptake of isoprene-derived epoxydiols (IEPOX) onto sulfate aerosol particles. In our previous studies, we have reported that high concentration of MTLs was related with sulfate, which could significantly promote the formation of isoprene-SOA tracers (Liu et al., 2020). Other studies also found that sulfate could increase the BSOA production by promoting acid-catalyzed ring-opening reactions (Xu et al., 2015). In contrast, positive correlations between BSOA tracers and aerosol pH in winter were observed, indicating that the formation of BSOA was predominantly enhanced by other factors, except for the aerosol acidity. The aerosol pH in winter was higher than those in summer, probably due to the influence of nitrate-dominated aerosols. Also, the aged aerosols

through long-range transport might result in the increase of BSOA tracers and aerosol pH.

In addition, positive correlation between BSOA tracers and LWC was observed (Table 1), probably attributed to the effects of the LWC on determining the peak buffer pH (pKa*). Zheng et al. (2020) reported that the buffering effect of ammonia suppresses the contribution of different chemical compositions in aerosol particles, making LWC the primary determinant of aerosol pH. Other studies have demonstrated that the uptake coefficient of first-generation oxidation products, especially for carbonyl compounds, might depend on RH (Luo et al., 2019). Meanwhile, high LWC could reduce the aerosol particle viscosity, which was benefit to the generation of the reactive intermediate such as IEPOX, or other oxidation products of VOC into aqueous-phase of aerosol particles, thereby promoting the formation of BSOA (Zhang et al., 2019b; Zhang et al., 2019d).

*3.5 Impacts of chlorine on BSOA formation*

Halogen radicals (Cl, Br and I) originated from sea salt aerosol (SSA) have an important role in tropospheric oxidants chemistry (Wang et al., 2021c). In this study, chlorine depletion was frequently observed in summer (Fig.7a), indicating that HCl can be formed through acid displacement of sea salt aerosol $Cl^-$ by $H_2SO_4$ and $HNO_3$ produced from anthropogenic emissions of $SO_2$ and NOx. Moreover, concentrations of the total SOA tracers were positively correlated with HCl (Fig.7a), suggesting the enhancement of SOA precursors transformation. Previous studies have found that Cl-initiated VOC oxidations could contribute to the formation of SOA (Wang and Ruiz, 2017; Dhulipala et al., 2019).

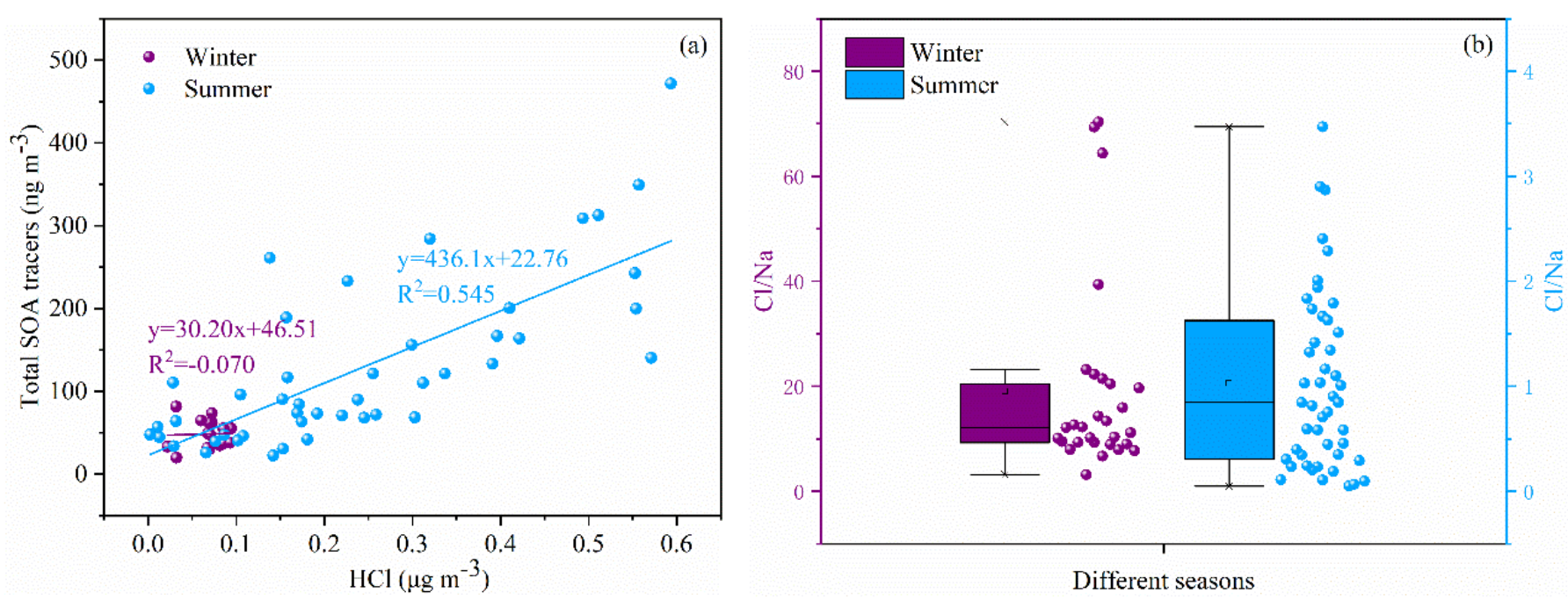

**Figure 7. Correlations of total SOA tracers and HCl (a) and chlorine depletion (b) in different seasons**

Under ammonia-rich conditions, HCl partitioned into the aqueous particulate phase mostly took place, and chlorine ions could affect aqueous oxidation of secondary organic compounds (Xu et al., 2021). As shown in Table 1, the correlations of SOA tracers in winter were found to increase with increasing $NH_3$ and chlorine ions in $PM_{2.5}$, while inverse results were observed in summer. In winter, the dominant wind direction is northeast (Fig.8), and chlorine mainly come from continental polluted air mass, such as industrial and combustion emissions. So, anthropogenic pollutants through long-range transport might cause the enhancement of SOA tracer concentrations at the monitoring site. However, in summer, negative correlations of BSOA tracers and chlorine ions in $PM_{2.5}$ was found, probably due to the influence of chlorine depletion. As shown in Fig. 8, the dominant wind direction is southerly, and chlorine mainly originated from the spray of sea salt.

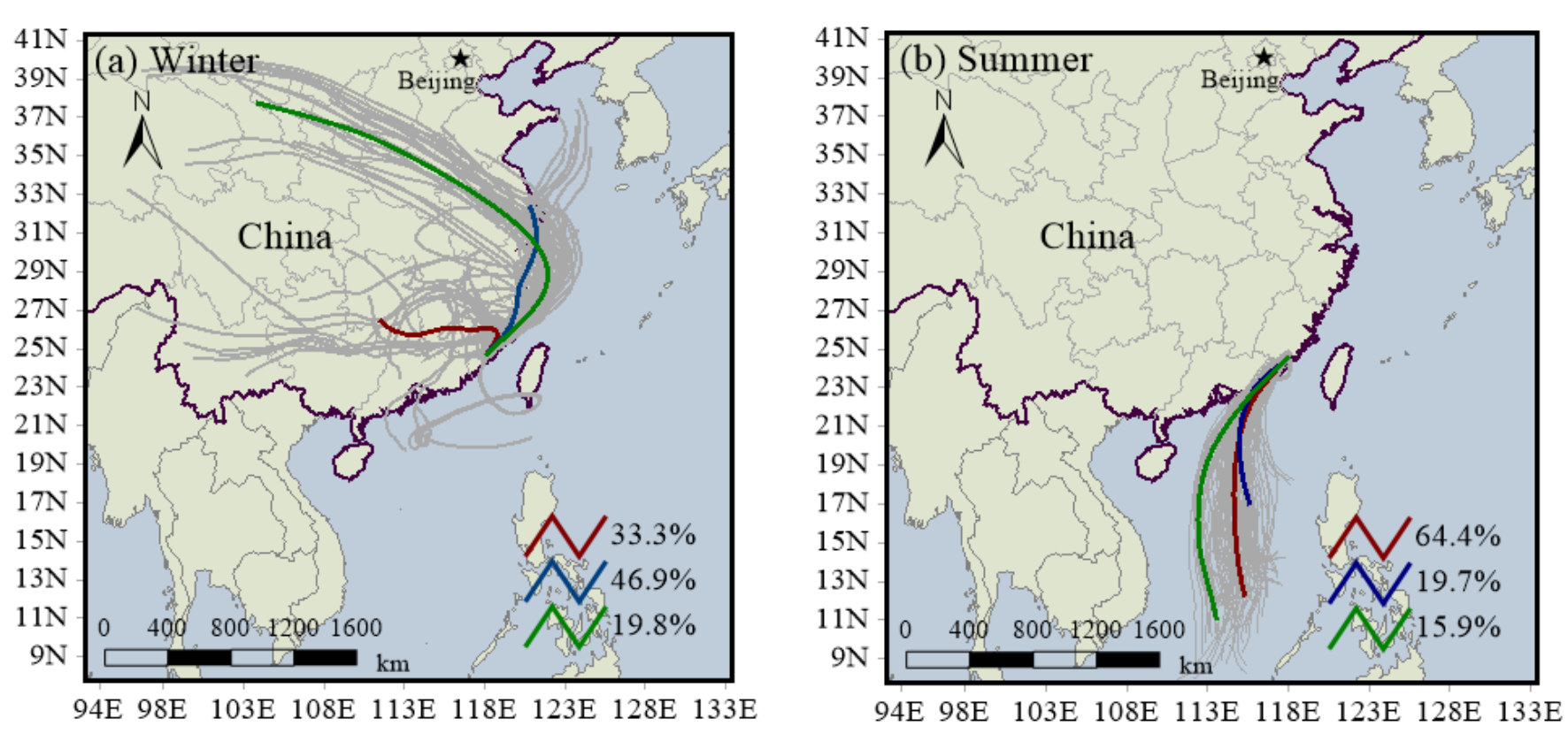

**Figure 8. Backward trajectories analyses during the winter (a) and summertime (b)**

*3.6. Enhanced formation of BSOA by anthropogenic emissions*

Recent studies had indicated that anthropogenic emissions might affect SOA formation through multiple chemical processes, based on laboratory studies and field

observations (Kari et al., 2019; Shrivastava et al., 2019; Zhang et al., 2019c; Cheng et al., 2021; Xu et al., 2021). In this study, we conducted the correlation analysis of individual SOA tracers and Ox(=$O_3$+$NO_2$), HONO, OH, $SO_2$, $NH_3$, $PM_{2.5}$, sulfate, nitrate, as well as meteorological parameters (including T, RH and UV) (Table 1).

Most of SOA tracers have a significant positive correlation with $NH_3$, suggesting an enhancement effect on the formation of SOA (Table 1). $NH_3$ could affect the SOA yields through both gas-phase and heterogeneous reactions (Na et al., 2007; Ma et al., 2018; Hao et al., 2020). Gas-phase reaction between $NH_3$ and organic acids (such as PA and PNA) produced ammonium salts in the particle phase, which contributed to the increased SOA formation. However, not all gas-phase organic acids (e.g., MGA and pyruvic acid) could demonstrate gas-to-particle conversion (Na et al., 2007). When SOA formation had ceased, the addition of excessive $NH_3$ would result in the rapid decomposition of the main SOA species, due to the nucleophilic attack of $NH_3$ (Ma et al., 2018).

**Table 1 Correlations between individual BSOA tracer and environmental factors in winter and summer**

| Season | SOA tracer | pH | LWC | HONO | $PM_{2.5}$ | Cl- | $NO_3$- | $SO_4^{2-}$ | $NH_3$ | $SO_2$ | $NO_2$ | Ox | T | RH | UV |
|---|---|---|---|---|---|---|---|---|---|---|---|---|---|---|---|
| WINTER (n=39) | C5 | .584** | .701** | .534** | .690** | .569** | .710** | .663** | .705** | 0.308 | .353* | 0.203 | .361* | 0.140 | 0.200 |
| | MTLs | .590** | .705** | .431* | .665** | .639** | .707** | .651** | .757** | 0.185 | 0.229 | 0.098 | .353* | 0.295 | -0.068 |
| | MGA | .390* | .707** | 0.261 | .668** | 0.081 | .758** | .572** | 0.284 | 0.172 | 0.123 | .374* | .377* | -0.019 | 0.238 |
| | PA | .432* | .403** | .463** | .407** | .481* | .416* | .488* | .440* | .446* | 0.241 | -0.193 | .319* | -0.205 | 0.145 |
| | PNA | .489** | .579** | 0.311 | .459** | .516** | .573** | .533** | .543** | 0.08 | 0.071 | -0.101 | 0.121 | .337* | -0.122 |
| | HGA | .443* | .829** | .352* | .834** | .600** | .847** | .754** | .641** | 0.275 | 0.299 | .451** | .451** | 0.043 | 0.210 |
| | MBTCA | .433* | .678** | .447** | .670** | .435* | .733** | .589** | .710** | .327* | 0.253 | .492** | .552** | -0.158 | 0.317 |
| | HDMGA | .421* | .876** | .401* | .867** | .631** | .884** | .813** | .643** | .335* | .321* | .526** | .485** | -0.049 | 0.327 |
| | AGA | .570** | .575** | .370* | .488** | .577** | .566** | .544** | .731** | 0.126 | 0.181 | 0.019 | 0.279 | 0.298 | -0.122 |
| | CPA | 0.212 | .462** | -0.068 | .452** | .483** | .437* | .419* | 0.255 | -0.15 | -0.170 | 0.016 | 0.079 | 0.200 | -0.144 |
| SUMMER (n=50) | C5 | -.495** | .425** | 0.160 | .622** | -.340* | 0.268 | .625** | .436** | 0.254 | 0.025 | .649** | .573** | -.529** | 0.247 |
| | MTLs | -.551** | 0.131 | 0.055 | 0.272 | -.439** | 0.131 | .428** | .304* | 0.089 | -0.278 | .550** | .610** | -.594** | 0.263 |
| | MGA | -.540** | 0.029 | 0.116 | 0.132 | -.403** | 0.066 | .472** | 0.270 | 0.096 | -.410** | .443** | .633** | -.668** | .382* |
| | PA | -.633** | .483** | .601** | .461** | -0.135 | .541** | .502** | .405* | 0.037 | 0.238 | .456** | .626** | -.558** | .400* |
| | PNA | -.664** | .616** | .387** | .812** | -.389** | .450** | .784** | .503** | 0.269 | .294* | .769** | .718** | -.631** | .404* |
| | HGA | -.607** | .612** | .299* | .836** | -.384** | .447** | .770** | .539** | .316* | 0.272 | .808** | .670** | -.599** | 0.322 |
| | MBTCA | -.752** | .415** | 0.237 | .577** | -.382** | .359* | .636** | .501** | 0.201 | -0.052 | .712** | .852** | -.816** | .588** |
| | HDMGA | -.525** | .618** | .299* | .833** | -.342* | .408** | .768** | .488** | .358* | .365** | .746** | .574** | -.500** | 0.240 |
| | AGA | -.684** | .592** | .447** | .766** | -.334* | .479** | .735** | .435** | 0.244 | 0.271 | .694** | .720** | -.634** | .477** |
| | CPA | -.552** | .625** | .441** | .780** | -.280* | .453** | .763** | .307* | .299* | .503** | .611** | .529** | -.458** | 0.305 |

*/**Correlation coefficients with an asterisk indicate statistically significant relationships at a = 0.05, and two asterisks mean significant at a = 0.01.

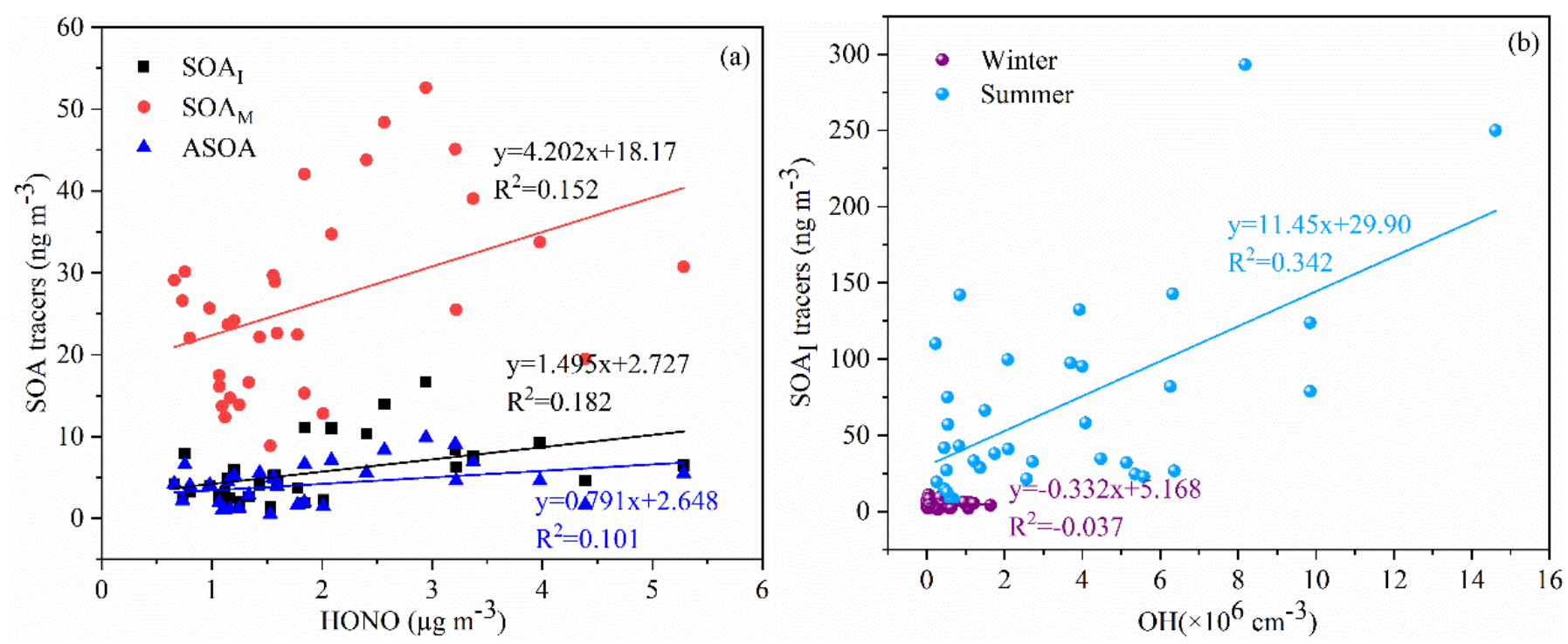

**Figure 9. Relationships of SOA tracers and HONO and its estimated OH**

As an indicator of atmospheric oxidation capacity, the tropospheric odd oxygen Ox ($O_3$+$NO_2$) was calculated. As shown in Table 1, the majority of SOA tracers in summer showed significant positive correlations with Ox (R>0.5, P<0.001). However, in winter, a part of SOA$_M$ tracers (e.g. HGA, MBTCA and HDMGA) were found to be significantly correlated with Ox. In addition, HONO and OH radicals, another critical indicator of atmospheric oxidation capacity, was also discussed. In this study, the concentration of OH radicals calculated from HONO in summer was higher than those in winter. In summer, the SOA$_I$ tracers was correlated with OH radicals (Fig.9b), consisted with previous findings that OH radicals could promote the formation of SOA (Sarrafzadeh et al., 2016; Liu et al., 2019; Song et al., 2019; Zhang et al., 2019a). Due to its photolysis to produce OH radicals during the daytime, HONO could facilitate SOA formation. In winter, the concentrations of SOA$_I$, SOA$_M$ and ASOA tracers were correlated with HONO (Fig.9a). These results indicated high concentrations of HONO and sufficient ultraviolet radiation could enhance the photochemical reactions of VOCs. Which was consisted with our previous results on the formation of peroxyacetyl nitrate (PAN) (Hu et al., 2020). As for T and UV, it exhibited significantly positive correlations with the related SOA tracers, especially in summer. These results suggested that SOA tracers were produced from the photo-oxidation of VOC precursors (Cheng et al., 2021).

In addition, the SOA tracers were significantly positive correlated with $PM_{2.5}$ and its components including $NO_3$- and $SO_4^{2-}$. In coastal cities of southeastern China, with

the development of rapid urbanization, air pollution caused by motor vehicles and industrial emissions is becoming more and more obvious in winter (Wu et al., 2020). Secondary formation of $PM_{2.5}$ accounted for 60-70% of the total fine particle, and $NO_3^-$, $SO_4^{2-}$ and $NH_4^+$ are significant components of secondary inorganic aerosols (Wu et al., 2019; Hong et al., 2021). These results also proved the obvious effects of anthropogenic emissions on secondary formation of aerosol particles under atmospheric relatively stability conditions during the winter.

## Conclusions

Pollution characteristics and source identification of BSOA tracers during the summer and winter in coastal areas of southeastern China were investigated. The average concentration of total BSOA tracers in summer was higher than that in winter, with the predominance of $SOA_M$, followed by $SOA_I$ and $SOA_C$. The BSOA tracers in summer were predominantly produced by the influence of photochemical oxidation under relatively clean conditions. However, in winter, the formation of BSOA tracers were attributed to the impacts of anthropogenic emissions, reflecting the anthropogenic–biogenic interactions. In addition, the results also indicated that acid-catalyzed reactive uptake onto sulfate aerosol particles enhanced the formation of BSOA in both seasons. We further found that Cl-initiated VOC oxidations has potentially accelerated the transformation of BSOA precursors through sea salt aerosol originated from the ocean in summer and anthropogenic emissions in winter. This study demonstrated that the combined effects of anthropogenic pollutants and atmospheric oxidation capacity on the formation of BSOA in coastal area.

***Data Availability.*** The data set related to this work can be accessed via https://doi.org/10.5281/zenodo.6376025 (Hong, 2022). The details are also available upon request from the corresponding author (ywhong@iue.ac.cn).

***Authorship Contribution Statement.*** Youwei Hong and Xinbei Xu contributed equally to this work. Youwei Hong designed and wrote the manuscript. Xinbei Xu collected the data, contributed to the data analysis. Dan Liao, Taotao Liu, Xiaoting Ji and Ke Xu performed modeling analyses and data analysis. Jinsheng Chen supported funding of observation and research. Chunyang Liao, Ting Wang and Chunshui Lin contributed to revise the manuscript.

***Competing interests.*** The authors declare that they have no conflict of interest.

***Acknowledgement.*** The authors gratefully acknowledge Yanting Chen, Han Zhang and Xu Liao (Institute of Urban Environment, Chinese Academy of Sciences) for the guidance and assistance during sample pretreatment, and Lingling Xu and Mengren Li (Institute of Urban Environment, Chinese Academy of Sciences) for the discussion of this paper.

***Financial support.*** This research was financially supported by the Xiamen Youth Innovation Fund Project (3502Z20206094), the foreign cooperation project of Fujian Province (2020I0038), the Cultivating Project of Strategic Priority Research Program of Chinese Academy of Sciences (XDPB1903), the National Key Research and Development Program (2016YFC0112200), State Key Laboratory of Environmental Chemistry and Ecotoxicology, Research Center for Eco-Environmental Sciences, CAS (KF2020-06), the FJIRSM&IUE Joint Research Fund (RHZX-2019-006) and center for Excellence in Regional Atmospheric Environment project (E0L1B20201).

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
