# Peer review of "Measurement Report: Effects of anthropogenic emissions and"

_Atmospheric Chemistry and Physics, 2022_

## Author Comment (AC1)

Editor and Reviewer comments:

Response: We thank the editor and reviewers for good comments and suggestions. We have addressed each comment in the following point by point. In addition, we have adjusted our reference list according to the ACP guideline.

**RC1: 'Comment on acp-2022-220', Anonymous Referee #1, 04 Apr 2022**

The work by Honga et al. investigated distribution of several organic tracer compounds, water-soluble inorganic ions in PM2.5 and gas phase HCl, HONO, HNO3, NH3 species in coastal areas of South-eastern China. The authors employed well established analytical techniques for identification and quantification of tracer compounds (e.g. TMS derivatisation). The obtained results are interesting and can be useful for researchers dealing with tracer compounds. I recommend this work for publication under *Measurments Reports* after considering my comments below:

Response: Thank you very much for all the valuable comments and suggestions. We have addressed each comment in the following point by point and have revised the manuscript accordingly.

**Materials and methods:**

The authors use **a single** internal standard (IS) to cover **fifteen** organic tracer compounds: Lines 152-153 state "At last, 140  $\mu$ L of internal standard solution (13 C n-alkane solution, 1.507 ng  $\mu$  L -1 ) was added into the samples". The majority of considered tracer compounds are of highly polar nature (containing hydroxylic groups). What was the rationale for selecting a non-polar 13C n-alkane as an IS for polar compounds? One of the requirements for IS that it should structurally resemble the analyte of interest (structural analogue or stable label) such that it behaves similarly during sample preparation and analysis (Lowes et al., 2011). The IS that is added to each sample compensates for unavoidable assay variance due to, for example, extraction efficiency, ionisation effects and transfer losses, and thus I am concerned about the discussion of correlation of various tracers in this work if the observed variability or absence of correlation could be due to other than environmental variability factors.

Response: Thank you for your kindly comments and good suggestions. We have described it clearly in the revised manuscript. In this study, four surrogate standards (structurally resemble the analytes of interest) was used to compensate for unavoidable assay variance in each sample during the pretreatment process, then internal standard (IS) was added after this process and before the instrument analysis.

Then, relative response factors (RRFs) of surrogate and internal standard were calculated to quantify the targeted organic compound in each sample, including SOAI, SOAM, SOAC and SOAA tracer.

These sentences have been rewritten, as follows:

Due to the lack of authentic standards, surrogate standards (including erythritol, malic acid, PA and citramalic acid) were used to compensate for unavoidable assay variance of SOAI, SOAM, SOAC and SOAA tracer in each sample during the pretreatment process, respectively (Fu et al., 2009; Lowes et al., 2011).

Then, relative response factors (RRFs) of surrogate and internal standard were calculated to quantify the targeted organic tracers in each sample. Details of SOA tracer's calculated concentrations based on RRFs were presented in our previous studies (Hong et al., 2019; Liu et al., 2020).

Lowes, S., Jersey, J., Shoup, R., Garofolo, F., Savoie, N., Mortz, E., Needham, S., Caturla, M. C., Steffen, R., Sheldon, C., Hayes, R., Samuels, T., Di Donato, L., Kamerud, J., Michael, S., Lin, Z. P., Hillier, J., Moussallie, M., Teixeira, L. D., Rocci, M., Buonarati, M., Truog, J., Hussain, S., Lundberg, R., Breau, A., Zhang, T. Y., Jonker, J., Berger, N., Gagnon-Carignan, S., Nehls, C., Nicholson, R., Hilhorst, M., Karnik, S., de Boer, T., Houghton, R., Smith, K., Cojocaru, L., Allen, M., Harter, T., Fatmi, S., Sayyarpour, F., Vija, J., Malone, M., and Heller, D.: Recommendations on: internal standard criteria, stability, incurred sample reanalysis and recent 483s by the Global CRO Council for Bioanalysis, Bioanalysis, 3, 1323-1332, 10.4155/Bio.11.135, 2011.

**Results and discussion:**

The authors give a fair description of isoprene oxidation products; however, I can't say the same about the other discussed tracers. For example, I realise that levoglucosan is commonly used as a marker compound for biomass burning; however, nothing is stated about stability of this compound. It has been shown that the oxidation of levoglucosan in atmospheric deliquescent particles is at least as fast as that of the other atmospherically relevant organic compounds and levoglucosan may not be as stable in the atmosphere, especially under high relative humidity conditions (Hoffmann et al., 2010). Can this be one of the reasons for absence of correlation with other tracers? Could you elaborate why are you expecting a correlation of CPA with levoglucoasan (lines 357-358)? This is not clear to me. As I understand, the applied derivatisation technique allows separation of other biomass burning markers e.g. mannosan and galactosan, which often accompany levoglucosan. Have the authors observed these isomers along with levoglucosan? The relative ratios of levoglucosan to mannosan have been used for source reconstruction of combustion derived byproducts in atmospheric aerosols (e.g. linuma et al., 2007, 2009, Engling et al., 2009) and can be useful to support some of the conclusions made in this work.

Response: Thank you for your good comments and suggestions. Indeed, as you mentioned, levoglucosan is commonly used as a marker compound for biomass burning, and may not be as stable in the atmosphere, especially under high relative humidity conditions. In this study, maybe, it's hard to reflect the real concentration of levoglucoasan, and we do not try demonstrate the variations and sources of levoglucosan during the monitoring period. So, we didn't talk about the stability of this compound. But, the seasonal and diurnal trend of levoglucoasan could be referred. A correlation of CPA with levoglucoasan was carried out to discuss the impacts of biomass burning on the distribution of SOA tracers through local or long-range transport. CPA, the typical tracer of sesquiterpenes, is formed by the photooxidation of  $\beta$ -caryophyllene. Some of them originated from the emission of biomass burning.

Due to the lack of authentic standards, mannosan and galactosan were not measured. The reviewer raised a good point. In the future, we will pay more attention to the characteristics of biomass burning markers including levoglucoasan, mannosan and galactosan when our researches focus on the effects of biomass burning on chemical compositions of aerosol particles.

In addition, DHOPA, an anthropogenic SOA tracer, was used to reflect the influence of anthropogenic activities emissions. Aromatic hydrocarbons (AHs) are typical AVOCs and a major class of ASOA precursors. In this study, the correlation between CPA and DHOPA was analyzed (Fig.S2) in order to discuss the influence of anthropogenic emissions on the source of CPA. We didn't comprehensively elaborate the variations and sources of DHOPA during the monitoring period.

These sentences have been added in the manuscript, as follows:

Levoglucosan (LEV), a typical tracer of biomass burning, similar seasonal and diurnal trend to other tracers was observed. However, LEV may not be as stable in the atmosphere, especially under high relative humidity conditions (Hoffmann et al., 2010). In this study, maybe, it's hard to reflect the real concentration of LEV. A correlation of CPA with LEV was carried out (Fig.S2), just to discuss the impacts of biomass burning on the distribution of CPA tracers through local or long-range transport.

Hoffmann, D., Tilgner, A., Iinuma, Y., and Herrmann, H.: Atmospheric Stability of Levoglucosan: A Detailed Laboratory and Modeling Study, Environmental Science & Technology, 44, 694-699, 10.1021/es902476f, 2010.

**Conclusion section:**

At least the way how it is formulated in the text I find it rather difficult to see how the presented work led to the conclusion that there is an impact from anthropogenic–biogenic interaction.

Response: Thank you for your suggestions. We have changed the description of "anthropogenic–biogenic interaction" in the revised manuscript.

The sentence has been rewritten, as follows:

However, in winter, the formation of BSOA tracers were attributed to the impacts of anthropogenic emissions and atmospheric stagnant conditions.

**Minor comment:**

Line 27 (page 23) The authors state "These results also proved the obvious effects of anthropogenic emissions on secondary formation of aerosol particles under atmospheric relatively stability conditions during the winter." I think the use of correlations is indeed helpful to support some specific trends; however, I believe such data processing techniques are not sufficient to provide a definite answer on the specific emission source and therefore the words such as "obvious" should be avoided (at least in this context), or supported by other than correlation data.

Response: Thank you for your suggestions. We have deleted the word "obvious" in the revised manuscript.

These sentences have been rewritten, as follows:

In coastal cities of southeastern China, with the development of rapid urbanization, air pollution caused by motor vehicles and industrial emissions is becoming more frequent in winter (Wu et al., 2020). The Xiamen port is one of the top 10 ports in China, resulting the impacts of ship emissions and port activities on ambient air quality (Xu et al., 2018), and the numbers of motor vehicles increased sharply in recent years. We also found that the 90th percentile of maximum daily average 8h (MDA8) O3 concentrations in Xiamen was significantly increased from 2015 to 2020 (Fig. S3). During the past several years, the elevated secondary inorganic components, including NO3-, SO42- and NH4+, accounted for 40-50% of the total PM2.5, and OM ranged from 30% to 40% (Wu et al., 2019; Hong et al., 2021). These results also implied the effects of anthropogenic emissions and enhanced atmospheric oxidation capacity on secondary formation of aerosol particles under atmospheric stagnant conditions.

Fig.S3 Annual trends of the 90th percentile MDA8 O3 concentrations in Xiamen

Xu, L., Jiao, L., Hong, Z., Zhang, Y., Du, W., Wu, X., Chen, Y., Deng, J., Hong, Y., and Chen, J.: Source identification of PM2.5 at a port and an adjacent urban site in a coastal city of China: Impact of ship emissions and port activities, Science of the Total Environment, 634, 1205-1213, 10.1016/j.scitotenv.2018.04.087, 2018.

---

## Author Comment (AC2)

**RC2**: ['Comment on acp-2022-220'](), Anonymous Referee #2, 18 Apr 2022

Review on "Measurement Report: Effects of anthropogenic emissions and environmental factors on biogenic secondary organic aerosol (BSOA) formation in a coastal city of Southeastern China" for Hong et al.

The author conducted the field observation during summer and winter in the southeast of China, and discussed the formation of SOA tracers, especially BSOA tracers. The author found that the concentrations of SOA tracers were affected by photochemical oxidation in summer, and were affected by anthropogenic emissions in winter. They highlighted that anthropogenic emissions, atmospheric oxidation capacity and halogen chemistry have significant effects on the formation of BSOA in the southeast coastal area. The manuscript can provide unique data for SOA tracers in the coastal area, and clarified the influencing factors on SOA formation. However, there are still some content deficiencies and logical omissions in this manuscript, which need to be carefully revised. Overall, the manuscript could be accepted after addressing the following issues.

Response: Thank you very much for all the valuable comments and suggestions. We have addressed each comment in the following point by point and have revised the manuscript accordingly.

1.  Line 147-149. How many times the samples were ultrasonically extracted during the pre-treatment, it should be shown in the manuscript.

Response: Thank you for your suggestions. The sentence was changed as follows:

Briefly, the filter samples were ultrasonically extracted with a mixture of dichloromethane and methanol (2:1, v/v) for 10 min three times.

2.  Line 189-190. $f_{SOC}$ of isoprene was 0.155 ± 0.039 in study of Kleindienst et al., 2007, the author should recheck your content.

Response: Thank you for your suggestions. Corrected.

3.  Section 2.5. The authors use both E-AIM IV model and ISORROPIA II model to calculate the aerosol pH. They need to discuss the correlation and difference between the results of two models, and explain which result is more reasonable for this manuscript. The authors should also explain which model they chose for the following discussions.

Response: Thank you for your good comments and suggestions. As the reviewer mentioned, E-AIM IV model and ISORROPIA II model are usually used to calculate the aerosol acidity. In this study, we compare them with each other. The comparison of $H^+$insitu calculated by EAIM IV and ISORROPIA II were illustrated in the

following figure. We found that the $H^+$insitu derived from ISORROPIA II agreed perfectly with those from E-AIM IV, and their trends matched perfectly with each other. For the two thermodynamic models, ISORROPIA II is widely used owing to its rigorous calculation, performance, and computational speed. Therefore, the results of ISORROPIA II calculation was just demonstrated in this study. To avoid the misunderstanding from the readers, we have deleted the introduction details of EAIM IV calculation.

[Figure]

Figure Comparison of $H^+$insitu calculated from E-AIM IV and ISORROPIA II.

The paragraph was rewritten as follows:

The forward mode of ISORROPIA II thermodynamic model was used to calculate the aerosol acidity (pH) (Fountoukis and Nenes, 2007). ISORROPIA II can calculate liquid water content (LWC), based on total $SO_4^{2-}$, $NO_3^-$, $Cl^-$, ammonia, non-volatile cations ($Na^+$, $K^+$, $Ca^{2+}$, $Mg^{2+}$), and meteorological factors (RH and T) (Rumsey et al., 2014; Guo et al., 2016). The pH value from ISORROPIA II was calculated using the following equation:

$$pH = - \lg\left(\frac{1000 \times H^+}{LWC}\right)$$

where $H^+$ is the hydronium ion concentration loading for an air sample ($\mu g/m^3$).

Fountoukis, C., and Nenes, A.: ISORROPIA II: a computationally efficient thermodynamic equilibrium model for $K^+$–$Ca^{2+}$–$Mg^{2+}$–$NH_4^+$ –$Na^+$–$SO_4^{2-}$ –$NO_3^-$–$Cl^-$–$H_2O$ aerosols, Atmos. Chem. Phys., 7, 4639-4659, 10.5194/acp-7-4639-2007, 2007.

Guo, H., Sullivan, A. P., Campuzano-Jost, P., Schroder, J. C., Lopez-Hilfiker, F. D., Dibb, J. E., Jimenez, J. L., Thornton, J. A., Brown, S. S., Nenes, A., and Weber, R. J.: Fine particle pH and the partitioning of nitric acid during winter in the northeastern United States, Journal of Geophysical Research: Atmospheres, 121, 10,355-310,376, https://doi.org/10.1002/2016JD025311, 2016.

Rumsey, I. C., Cowen, K. A., Walker, J. T., Kelly, T. J., Hanft, E. A., Mishoe, K., Rogers, C., Proost, R., Beachley, G. M., Lear, G., Frelink, T., and Otjes, R. P.: An assessment of the performance of the Monitor for AeRosols and GAses in ambient air (MARGA): a semi-continuous method for soluble compounds, Atmos. Chem. Phys., 14, 5639-5658, 10.5194/acp-14-5639-2014, 2014.

4. Section 3.1. In my opinion, it is clearer to list the average concentrations of these air pollutants during summer and winter, daytime and nighttime in Supporting Information as a Table.

Response: Thank you for your good suggestions. The details have been shown in Table S1.

Table S1 Comparisons of criteria air pollutants and meteorological parameters during the daytime and nighttime in winter and summer

| Index | Winter | | Summer | |
|---|---|---|---|---|
| | Daytime | Nighttime | Daytime | Nighttime |
| $PM_{2.5}(\mu g/m^3)$ | 40.3±18.7 | 45.1±17.0 | 19.4±9.70 | 14.1±6.00 |
| $PM_{10}(\mu g/m^3)$ | 61.1±27.2 | 68.9±25.0 | 36.5±17.5 | 30.3±9.70 |
| $O_3(\mu g/m^3)$ | 45.7±25.4 | 37.6±16.8 | 80.3±46.2 | 24.2±11.8 |
| $CO(mg/m^3)$ | 0.70±0.10 | 0.70±0.10 | 0.30±0.10 | 0.30±0.10 |
| $SO_2(\mu g/m^3)$ | 2.90±1.80 | 2.10±0.90 | 8.30±1.00 | 7.80±1.40 |
| $NO_2(\mu g/m^3)$ | 33.0±8.50 | 32.3±9.00 | 12.2±6.50 | 18.7±7.40 |
| T(℃) | 16.8±2.60 | 14.6±1.70 | 36.0±2.70 | 31.2±1.00 |
| P(kPa) | 100.9±0.20 | 100.9±0.20 | 99.5±0.20 | 99.6±0.20 |
| RH(%) | 60.7±9.50 | 69.5±5.80 | 55.0±6.90 | 67.7±3.30 |
| WD(°) | 159.0±14.3 | 151.3±12.7 | 191.5±16.9 | 194.0±30.8 |
| WS(m/s) | 1.50±0.40 | 1.10±0.70 | 1.40±0.30 | 0.80±0.20 |

5. Line 250. The average concentrations of $SOA_M$, $SOA_I$ and $SOA_C$ in winter and summer should be given. As the author determined to discuss "total SOA tracers" (Line 249), the concentration of ASOA should also be shown here.

Response: Thank you for your good comments. These sentences have been added in the revised manuscript, as follows:

The average concentrations of total SOA tracers in winter and summer were 37.3 and 111.3 ng m$^{-3}$, respectively. The predominance of SOA$_M$ (26.6 ng m$^{-3}$), followed by ASOA (4.60 ng m$^{-3}$), SOA$_I$ (4.35 ng m$^{-3}$) and SOA$_C$ (1.76 ng m$^{-3}$) was observed in winter while SOA$_I$ (54.4 ng m$^{-3}$) and SOA$_M$ (47.8 ng m$^{-3}$) in summer were the main contributors to total SOA tracers, followed by ASOA (6.64 ng m$^{-3}$) and SOA$_C$ (2.45 ng m$^{-3}$).

6. Line 250-252. The author showed that "In summer, BSOA tracers showed much higher concentrations in the daytime than in the nighttime, while inverse results were observed in winter", the specific concentrations of BSOA tracers in daytime and nighttime of summer and winter should be displayed here.

Response: Thank you for your suggestions. Corrected.

In summer, BSOA tracers showed much higher concentrations in the daytime (149.3 ng m$^{-3}$) than in the nighttime (60.1 ng m$^{-3}$), while inverse results were observed in winter (30.4 ng m$^{-3}$ and 35.0 ng m$^{-3}$ in the daytime and nighttime, respectively)

7. Line 252-258. Instead of using "for example" here, the author could display the average concentrations of SOA tracers (including SOA$_I$, SOA$_M$, SOA$_C$ and ASOA tracers) during day, night, summer and winter in the Supporting Information as a Table directly.

Response: Thank you for your good suggestions. The details have been shown in Table S2.

Table S2 Comparisons of different types of SOA tracers (ng m$^{-3}$) during the daytime and nighttime in winter and summer

| SOA tracers | Winter | | Summer | |
|---|---|---|---|---|
| | Daytime | Nighttime | Daytime | Nighttime |
| SOA$_I$ | 3.79±2.37 | 4.91±3.75 | 81.9±66.2 | 26.8±24.8 |
| SOA$_M$ | 24.9±8.51 | 28.3±13.0 | 64.5±38.5 | 31.2±27.2 |
| SOA$_C$ | 1.70±0.81 | 1.82±0.77 | 2.83±1.97 | 2.06±2.11 |
| Sum of BSOA | 30.4±11.1 | 35.0±17.1 | 149.3±96.9 | 60.1±52.9 |
| ASOA | 3.80±1.99 | 5.35±2.72 | 9.00±5.98 | 4.28±2.96 |
| Total SOA | 34.2±12.8 | 40.4±19.6 | 158.3±102.5 | 64.4±55.8 |

8. Line 275-279. As the concentrations of SOA tracers were higher in summer than winter, and the f$_{SOC}$ values were constant in this manuscript, it was not surprisingly that the concentrations of SOC in summer was higher than that in winter. And this result could not demonstrate that the contributions of SOA tracers to SOC in summer was higher than those in winter.

Response: Thank you for your comments. The sentence has been revised as follows:

The concentrations of SOC in summer was higher than that in winter, attributed to the increase of flourishing vegetation emissions and photochemical reactions under high temperature and strong solar radiation conditions.

9. Line 283-286. This sentence is confusing, why does the "obvious trend of diurnal variations of SOC$_I$" was "consistent with the isoprene emission", and why this result was compared with the trend in winter? Considering the coherence of context, maybe the author intended to explain the diurnal variation of SOC$_I$ was obvious in summer and the variation was consistent with isoprene emission in summer? The authors should give more explanation about it.

Response: Thank you for your kindly comments. Exactly, as the reviewer mentioned, we try to demonstrate the diurnal variation of SOC$_I$ was obvious in summer and the variation was consistent with isoprene emission in summer. We analyze the diurnal variation of isoprene concentrations during the wintertime and summertime, as shown in Fig.S3. These sentences have been rewritten in the revised manuscript, as follows:

An obvious trend of diurnal variations of isoprene-derived SOC in summer was observed, which was consistent with the diurnal pattern of isoprene concentration (Fig.S3). However, no similar trend was found in winter, attributed to the influence of low temperature on inhibiting the emissions of isoprene from various kinds of plants.

[Figure]

Fig.S3. Diurnal variation of isoprene concentrations during the wintertime and summertime

10. Figure 3. The legend of Figure 3 might be $SOC_I$, $SOC_M$, $SOC_C$ and ASOC.

Response: Thank you for your kindly comments. Corrected.

11. Line 306, it should be "$SOA_I$ tracers", and Line 308, it should be "$SOA_M$ tracers".

Response: Corrected.

12. Line 319. I think the first (PA and PNA) and later generation (HGA, AGA, HDMGA and MBTCA) products could only evaluate the aging degree of $SOA_M$, not all BSOA.

Response: Thank you for your comments. The sentence has been rewritten as follows:

The first (PA and PNA) and later generation (HGA, AGA, HDMGA and MBTCA) products were used to evaluate the aging degree of $SOA_M$.

13. Line 333-335. According to the logic of this section, it might be "Low ratio of HGA/MBTCA (~1.0) showed that α-pinene was the major precursor for $SOA_M$. The ratio of HGA/MBTCA with an average of 5.78 in Xiamen was high, suggesting the contribution of β-pinene to $SOA_M$".

Response: Thank you for your comments. Corrected.

14. Line 362. The author used the pH values calculated by ISORROPIA II here. Same as the Q3, the author should explain why they chose the pH calculated by ISORROPIA II, but not that calculated by E-AIM IV.

Response: Thank you for your kindly comments. As mentioned in Q3, for the two thermodynamic models, ISORROPIA II is widely used owing to its rigorous calculation, performance, and computational speed.

15. Line 380. Table 1 should be listed after this paragraph, which refers to table 1 for the first time.

Response: Thank you for your comments. Corrected.

16. As the contents of Figure 6 and Table 1 are similar, and the author has not discussed Figure 6 in detail, this figure should be moved to the supporting information section.

Response: Thank you for your kindly suggestions. Figure 6 was moved to the SI section, named Fig.S4.

17. Line 425-427. The author showed that "the correlations of SOA tracers in winter were found to increase with increasing $NH_3$ and chlorine ions in $PM_{2.5}$, while inverse results were observed in summer". The sentence is not rigorous, because $NH_3$ was not negative correlated with SOA tracers in summer as shown in Table 1.

Response: Thank you for your good comments. The correlations between SOA tracers and $NH_3$ was discussed in 3.6. The sentence has been rewritten as follows:

As shown in Table 1, most of SOA tracers in winter were correlated with the concentrations of chlorine ions in $PM_{2.5}$, while inverse results were observed in summer.

---

## Author Comment (AC3)

**Comment on acp-2022-220** Anonymous Referee #3

Referee comment on "Measurement Report: Effects of anthropogenic emissions and environmental factors on biogenic secondary organic aerosol (BSOA) formation in a coastal city of Southeastern China" by Youwei Hong et al., Atmos. Chem. Phys. Discuss., https://doi.org/10.5194/acp-2022-220-RC3, 2022

I think this is a good submission to ACPD along the current line of thinking in atmospheric chemistry. The authors investigated ambient $PM_{2.5}$ in coastal areas of South-eastern China and reported experimental distribution of the main organic tracers (mainly BSOA), water-soluble inorganic ions and gas phase species including HCl, HONO, $HNO_3$, $NH_3$. The analytical method (qualitative and quantitative) used by Honga et al. is well established for these oxygenated compounds. The results of this study show that the concentrations associated with SOA organic tracers depends on the photochemistry in summer, and on
the emission of anthropogenic compounds in winter. The results of this study are interesting to the scientific community including modeling as it provides experimental link between photochemistry, anthropogenic emission and BSOA tracers in a coastal area of southeastern China. This work would be beneficial for publication under *Measurments Reports* after considering my comments below:

Response: Thank you very much for all the valuable comments and suggestions. We have addressed each comment in the following point by point and have revised the manuscript accordingly.

The analytical technique used IS and the authors should comment on the use of only one non-polar IS. I do recognize the difficulties of finding the correct IS due to co-elution issue with the number of oxygenated species that are detected in ambient $PM_{2.5}$. Ketopinic acid is used by several groups as IS as it could not be detected in ambient PM and is a polar oxygenated specie!!

Response: Thank you for your kindly comments and suggestions. We have described it clearly in the revised manuscript. In this study, four surrogate standards (structurally resemble the analytes of interest) was used to compensate for unavoidable assay variance in each sample during the pretreatment process, then internal standard (IS) was added after this process and before the instrument analysis. Then, relative response factors (RRFs) of surrogate and internal standard were calculated to quantify the targeted organic compound in each sample, including $SOA_I$, $SOA_M$, $SOA_C$ and $SOA_A$ tracer.

These sentences have been rewritten, as follows:

Due to the lack of authentic standards, surrogate standards (including erythritol, malic acid, PA and citramalic acid) were used to compensate for unavoidable assay variance of $SOA_I$, $SOA_M$, $SOA_C$ and $SOA_A$ tracer in each sample during the pretreatment process, respectively (Fu et al., 2009; Lowes et al., 2011).

Then, relative response factors (RRFs) of surrogate and internal standard were calculated to quantify the targeted organic tracers in each sample. Details of SOA tracer's calculated concentrations based on RRFs were presented in our previous studies (Hong et al., 2019; Liu et al., 2020).

Lowes, S., Jersey, J., Shoup, R., Garofolo, F., Savoie, N., Mortz, E., Needham, S., Caturla, M. C., Steffen, R., Sheldon, C., Hayes, R., Samuels, T., Di Donato, L., Kamerud, J., Michael, S., Lin, Z. P., Hillier, J., Moussallie, M., Teixeira, L. D., Rocci, M., Buonarati, M., Truog, J., Hussain, S., Lundberg, R., Breau, A., Zhang, T. Y., Jonker, J., Berger, N., Gagnon-Carignan, S., Nehls, C., Nicholson, R., Hilhorst, M., Karnik, S., de Boer, T., Houghton, R., Smith, K., Cojocaru, L., Allen, M., Harter, T., Fatmi, S., Sayyarpour, F., Vija, J., Malone, M., and Heller, D.: Recommendations on: internal standard criteria, stability, incurred sample reanalysis and recent 483s by the Global CRO Council for Bioanalysis, Bioanalysis, 3, 1323-1332, 10.4155/Bio.11.135, 2011.

Are additional compounds associated with isoprene detected (hydro-carboxylic acids)?
Response: Unfortunately, hydro-carboxylic acids was not measured in this study. The reviewer raised a good point. In the future, we will pay more attention to the pollution characteristics of hydro-carboxylic acids, beneficial to study the atmospheric chemistry process of SOA formation.

The authors should provide additional evidence from the present work on the interaction biogenic-anthropogenic and its effect on PM formation.

Response: Thank you for your good suggestions. Indeed, I think it rather difficult to see how the presented work led to the conclusion that there is an impact from anthropogenic–biogenic interaction. We have changed the description of "anthropogenic–biogenic interaction" in the revised manuscript.

The sentence has been rewritten, as follows:

However, in winter, the formation of BSOA tracers were attributed to the impacts of anthropogenic emissions and atmospheric stagnant conditions.

---

## Author Response (AR2)

**Comments to the author**:

Dear Authors,

Thank you for addressing the 3 reviewer comments. Upon reviewing your replies and your new version of the manuscript, I will now accept with minor revisions noted. Please reply point-by-point to these remaining comments and outline how you changed the manuscript as a result.

Thanks so much, Jason Surratt

Response: Thank you so much for your kindly comments and good suggestions. We have addressed each comment in the following point by point and revised the manuscript accordingly. In addition, we check our manuscript again, according to the ACP guideline.

1.) What are the uncertainties of using GC/MS and the SOC tracer method to estimate total SOC? This is the largest issue to me that isn't addressed in this manuscript. For example, recent studies have raised serious questions about how thermal methods like GC/MS might cause for misinterpretation of isoprene SOA (e.g., Cui et al., 2018, ESPI; Lopez-Hilfiker et al., 2016, ES&T). Specifically, these prior studies show that isoprene SOA has a very low volatile nature, which is inconsistent with chemical constituents like C5-alkene triols and even to some degree 2-methyltetrols (e.g. Lopez-Hilfiker et al., 2016, ES&T, Hu et al., 2016, ACP). I think the authors need to acknowledge at least for isoprene SOA (but this is likely true for other SOA types) that low-volatility oligomers (including some organosulfates) may break down into monomers like C5-alkene triols and 2-methyltetrols. The SOC estimates made in this study likely have a large degree of uncertainty due to the thermal breakdown of the "actual" SOA constituents and also due to the lack of more complete anthropogenic SOA tracers. I think the authors need to acknowledge these uncertainties for their study before publication can be fully considered. Lastly, the foc values described in Kleindienst et al. that are used in this study to estimate the SOA amounts has a lot of uncertainty, since these were determined for these tracers of ONE representative experimental scenario. I worry that this SOA tracer method gets overused and not properly acknowledged for its high degree of uncertainty.

Response: Thank you very much for your good comments and suggestions. You raised a very good point. Indeed, due to their inherent low volatility of isoprene SOA tracers, and more importantly a lack of knowledge about their identity and thus available authentic standards, quantifying the abundance of such accretion products has remained an analytical challenge. Nowadays, in our observations sites, online

chemical characterization of SOA was performed using a chemical ionization mass spectrometer (CIMS) equipped with a filter inlet for gases and aerosols (FIGAERO) and aerosol mass spectrometer (AMS-Aerodyne Research Inc.).

Some details of the uncertainties of using GC/MS has been added in the revised manuscript, as follows:

However, inherent low volatility of isoprene SOA tracers could cause the uncertainties of using the GC/MS method, and low-volatility oligomers might break down into monomers, such as C5-alkene triols and 2-methyltetrols (Lopez-Hilfiker et al., 2016; Hu et al., 2016). Therefore, quantifying the abundance of certain SOA tracers remained a lot of uncertainties.

Hu, W., Palm, B. B., Day, D. A., Campuzano-Jost, P., Krechmer, J. E., Peng, Z., de Sa, S. S., Martin, S. T., Alexander, M. L., Baumann, K., Hacker, L., Kiendler-Scharr, A., Koss, A. R., de Gouw, J. A., Goldstein, A. H., Seco, R., Sjostedt, S. J., Park, J.-H., Guenther, A. B., Kim, S., Canonaco, F., Prevot, A. S. H., Brune, W. H., and Jimenez, J. L.: Volatility and lifetime against OH heterogeneous reaction of ambient isoprene-epoxydiols-derived secondary organic aerosol (IEPOX-SOA), Atmospheric Chemistry and Physics, 16, 11563-11580, 10.5194/acp-16-11563-2016, 2016.

Lopez-Hilfiker, F. D., Mohr, C., D'Ambro, E. L., Lutz, A., Riedel, T. P., Gaston, C. J., Iyer, S., Zhang, Z., Gold, A., Surratt, J. D., Lee, B. H., Kurten, T., Hu, W. W., Jimenez, J., Hallquist, M., and Thornton, J. A.: Molecular Composition and Volatility of Organic Aerosol in the Southeastern US: Implications for IEPDX Derived SOA, Environmental Science & Technology, 50, 2200-2209, 10.1021/acs.est.5b04769, 2016.

In addition, the foc values in this study cited from Kleindienst et al. based on simulation experiments, also has a lot of uncertainty. Therefore, the results of SOC tracer method used to estimate total SOC have been simplified and moved to the supporting information section. The trends and percentages of different types of SOC were only demonstrated in this study.

2.) Line 44: Change "Compare" to "Compared"
   Response: Corrected.

3.) Lines 44-46: I would insert the result of your regression analyses in the abstract for the BSOA tracers versus Ox, HONO, UV, and T. Readers will want to assess from themselves how well correlated are these parameters.

Response: Thank you for your suggestions. The sentence has been rewritten, as follows:

Compared to those in winter, the majority of BSOA tracers in summer showed significant positive correlations with Ox (O$_3$+NO$_2$) (r = 0.443～0.808), HONO (r = 0.299～0.601), ultraviolet (UV) (r = 0.382～0.588) and temperature (T) (r = 0.529～0.852), indicating the influence of photochemical oxidation under relatively clean conditions.

4.) Lines 47-49: I would insert the result of your regression analyses in the abstract for the BSOA tracers versus PM2.5, NO3-, SO42-, and NH3. Readers will want to assess from themselves how well correlated are these parameters.
Response: Thank you for your suggestions. The sentence has been rewritten, as follows:

However, in winter, BSOA tracers were significantly correlated with PM$_{2.5}$ (r = 0.407～0.867), NO$_3^-$ (r = 0.416～0.884), SO$_4^{2-}$ (r = 0.419～0.813), and NH$_3$ (r = 0.440～0.757), attributed to the contributions of anthropogenic emissions.

5.) Lines 49-50: I would insert the result of your regression analyses in the abstract for the BSOA tracers versus aerosol acidity, LWC, and SO42-. Readers will want to assess from themselves how well correlated are these parameters.
Response: Thank you for your suggestions. The sentence has been rewritten, as follows:

Major BSOA tracers in both seasons was linearly correlated with aerosol acidity (pH) (r = 0.421～0.752), liquid water content (LWC) (r = 0.403～0.876) and SO$_4^{2-}$ (r = 0.419～0.813).

6.) Lines 53-54: I would insert the result of your regression analyses in the abstract for the SOA tracers versus HCl and Cl- ions in PM2.5. Readers will want to assess from themselves how correlated are these parameters.
Response: Thank you for your suggestions. The sentence has been rewritten, as follows:

We also found that concentrations of the total SOA tracers was correlated with HCl (R$^2$ = 0.545) and chlorine ions (r = 0.280～0.639) in PM$_{2.5}$, reflecting the contribution of Cl-initiated VOC oxidations to the formation of SOA.

7.) Lines 64-65: change "researchers" to "research"
    Response: Corrected.

8.) Line 73: Change "was" to "is"
    Response: Corrected.

9.) Line 78: Delete "observation" and change "model" to "modeling'
    Response: Corrected.

10.) Line 88: Change "nitrates" to "nitrate formation"
    Response: Corrected.

11.) Lines 114-115: Please rewrite the sentence "We also demonstrated the indications of SOA tracers for air pollution process." This sentence is not well-written and is unclear what it adds here to the introduction.

Response: Thank you for your suggestions. The sentence has been rewritten, as follows:

Atmospheric process identified by SOA tracers in different seasons were further analyzed.

12.) Looking at your figure comparing H+ insitu derived from ISORROPIA and EIAM IV it is hard to see they agreed perfectly. How did the authors conclude this? Did they from a linear correlation, and if so, what was the result of that correlation? I raise this issue as there are several time periods where EAIM calculates higher H+ insitu. Seeing this difference at many time periods raising the question for me as to which model is the best to use?
Response: Thank you for your good suggestions and comments. At first, both ISORROPIA and EIAM IV were tried to calculate the aerosol acidity in this study. But, different data were input into these two models. ISORROPIA II calculated the equilibrium $H_{air}^+$ and aerosol liquid water content of inorganic material ( ) by inputting the concentrations of the total $SO_4^{2-}$ ($TH_2SO_4$, replaced by observed $SO_4^{2-}$), total $NO_3^-$ ($TNO_3$, gas $HNO_3$ plus particle $NO_3^-$), total ammonia (NHx, gas $NH_3$ plus particle $NH_4^+$), total $Cl^-$ (TCl, replaced by observed $Cl^-$ due to the low concentration and measurement uncertainties of HCl)(Rumsey et al., 2014). However, the related air pollutants (including gas $HNO_3$ and $NH_3$) were not used in the EIAM IV.

Indeed, as you mentioned, there are several time periods where EAIM calculates higher H+ insitu. I think it has a lot of uncertainty, due to the 10-20 day observation period and different input data set. I regret not to answer the issue in this article. Anyway, the editor raised a very good point. In other study, we are comparing the difference of these two models based on multi-year monitoring data. We hope to find the answer in the future. Thank you for your good advice.

The sentence has been rewritten in the revised manuscript, as follows:

ISORROPIA II can calculate liquid water content (LWC), based on total $SO_4^{2-}$, $NO_3^-$ (gas $HNO_3$ plus particle $NO_3^-$), $Cl^-$, ammonia (gas $NH_3$ plus particle $NH_4^+$), non-volatile cations ($Na^+$, $K^+$, $Ca^{2+}$, $Mg^{2+}$), and meteorological factors (RH and T).